# Ancient dolphin genomes reveal rapid repeated adaptation to coastal waters

Marie Louis [1,2,3,4,20] ✉, Petra Korlević [5,6,20], Milaja Nykänen[7,8,20], Frederick Archer [9], Simon Berrow[10,11], Andrew Brownlow [12], Eline D. Lorenzen[2], Joanne O'Brien[10,11], Klaas Post[13], Fernando Racimo[2], Emer Rogan[8], Patricia E. Rosel [14], Mikkel-Holger S. Sinding [15], Henry van der Es[13], Nathan Wales [16], Michael C. Fontaine [3,17,21], Oscar E. Gaggiotti [1,21] & Andrew D. Foote [18,19,21] ✉

Parallel evolution provides strong evidence of adaptation by natural selection due to local environmental variation. Yet, the chronology, and mode of the process of parallel evolution remains debated. Here, we harness the temporal resolution of paleogenomics to address these long-standing questions, by comparing genomes originating from the mid-Holocene (8610-5626 years before present, BP) to contemporary pairs of coastal-pelagic ecotypes of bottlenose dolphin. We find that the affinity of ancient samples to coastal populations increases as the age of the samples decreases. We assess the youngest genome (5626 years BP) at sites previously inferred to be under parallel selection to coastal habitats and find it contained coastal-associated genotypes. Thus, coastal-associated variants rose to detectable frequencies close to the emergence of coastal habitat. Admixture graph analyses reveal a reticulate evolutionary history between pelagic and coastal populations, sharing standing genetic variation that facilitated rapid adaptation to newly emerged coastal habitats.

Parallel adaptation can arise from selection repeatedly acting upon standing genetic variation[1,2]. However, the timing and independence of selection is still poorly understood for most natural study systems[3,4]. For example, selection can act upon standing genetic variation independently in multiple derived populations, representing parallel adaptation. It can also act upon standing genetic variation in the ancestral population(s), which then splits into different populations, producing a pattern that can be wrongly inferred as parallel adaptation. Alternatively, standing genetic variation can be shared among derived populations through gene flow, before (i.e. parallel) or after selection (i.e. not parallel)[5–9].

In this study, we investigate the temporal dynamics and independence of adaptation to coastal habitat from standing genetic variation in common bottlenose dolphins (*Tursiops truncatus*) derived from pelagic ancestors using contemporary and ancient (8610–5626 years before present (BP)) genomes. Coastal and pelagic ecotypes of

bottlenose dolphins have recurrently formed in different regions of the world[10–13]. Coastal populations are thought to have been founded by pelagic individuals, and show lower genetic diversity and smaller effective population size than pelagic populations[10–15]. Morphological differences are found between the two ecotypes but are not consistent across geographical regions[12,13,16,17]. The coastal ecotype shares behavioural traits across its range such as restricted dispersal, high site fidelity and socially transmitted foraging techniques[10,18]. Single nucleotide polymorphisms (SNPs), also called variants hereafter, were found to be under parallel selection in geographically distant coastal habitats, that is under both homogenising selection among coastal populations from different ocean basins and divergent selection between coastal and pelagic ecotypes within each ocean basin. We refer to this process as parallel linked selection, as these previous analyses may have identified the targets of selection and/or loci physically linked to these targets[19]. The coastal-associated variants are

found mainly in ancient ancestry tracts in the genomes of coastal dolphins; these genomic regions have a time to the most recent common ancestor between coastal and pelagic counterparts of 0.6 to 2.3 million years BP instead of the genome-wide mean of 0.1 to 0.4 million years BP. Coastal-associated variants are present at low frequency as standing variation in pelagic populations[19]. However, the mode and chronology of parallel linked selection on coastal-associated genetic variation in bottlenose dolphins remain unresolved. Here, we explore potential scenarios, incorporating ancient genomes dating from the estimated time of formation of local coastal populations[12,20].

Four subfossil samples dredged from the southern North Sea bed in the eastern North Atlantic (ENA) and curated at the Natural History Museum Rotterdam, were radiocarbon-dated to 5979–5626 calendar years BP for the youngest sample (SP1060, 95% CI) and 8610–8243 years BP for the oldest sample (NMR10326, Supplementary Table 1, Fig. 1a, b and Supplementary Data 1). These ages fall within the 95% CI range of the estimated split time of pelagic and coastal ecotypes in the ENA (47,800–4300 years BP[12]) and the emergence of coastal habitat in northern European waters[20]. This coastal habitat emerged when Doggerland was submerged by the North Sea around 8000 to 6000 years BP following post-glacial sea-level rise[21]. During this warmer period of the early to mid-Holocene, subfossil evidence from the region confirms common bottlenose dolphin occurrence in the North Sea[22,23]. It is thought this species entered the North Sea through the English Channel after it opened up following deglaciation[22]. The emergence of new coastal habitats, with different abiotic and biotic environmental conditions such as salinity, depth and prey species, may have affected the physiology, foraging strategies and behaviour of bottlenose dolphins and created opportunities for local adaptation to occur.

In this study, to understand the importance of these four early- to mid-Holocene ancient samples in the chronology of coastal adaptation, we first established their relationship to contemporary dolphins. We sequenced an ancient dolphin genome, SP1060, at 3x effective coverage (i.e. post quality control filtering, repeat masking, removing

duplicates, base quality recalibration), to compare with a dataset of 60 contemporary genomes[19]. We sequenced an additional three ancient dolphin genomes at ultra-low effective coverage (<0.05x; Supplementary Table 1) to verify whether SP1060 was representative of the genetic variation within the ancient population, rather than, for example, a rare admixed individual. We show that parallel adaptation occurred rapidly after the emergence of new coastal habitats. We also provide insights into how past admixture retained coastal-adapted ancestry at low frequency in pelagic populations, which selection can then act upon to promote rapid adaptation.

## Results and discussion

### Relationship between ancient and contemporary dolphin genomes

To explore the relationship between the ancient subfossils and contemporary individuals, we first looked at mitochondrial ancestry. The mitogenome sequence of the oldest ancient sample NMR10326 (8610–8243 years BP, Fig. 1b) clustered with contemporary North Atlantic pelagic haplotypes, while the mitogenome sequences of the three other ancient samples clustered with contemporary Mediterranean and the Black Sea samples in a Bayesian phylogenetic analysis (Supplementary Fig. 2). Then, with the nuclear data, we used principal component analysis[24,25] (PCA) and a factor analysis[26] (FA) method that can incorporate sample age, and therefore allows us to correct for temporal genetic drift when comparing populations[26]. Given the differences in depth of coverage and the potential biases inherent in mapping and comparing ancient and contemporary genomes[27], we ran the multivariate analyses using several approaches: (i) comparing pseudo-haploid genomes generated by sampling a random single base for all genomic positions and using a PC projection of the ancient genomes on to the principal components segregating the contemporary genomes; (ii) sampling a single read for each site with no projection but including sites covered in all contemporary and at least one ancient sample, and (iii) comparing called genotypes for the

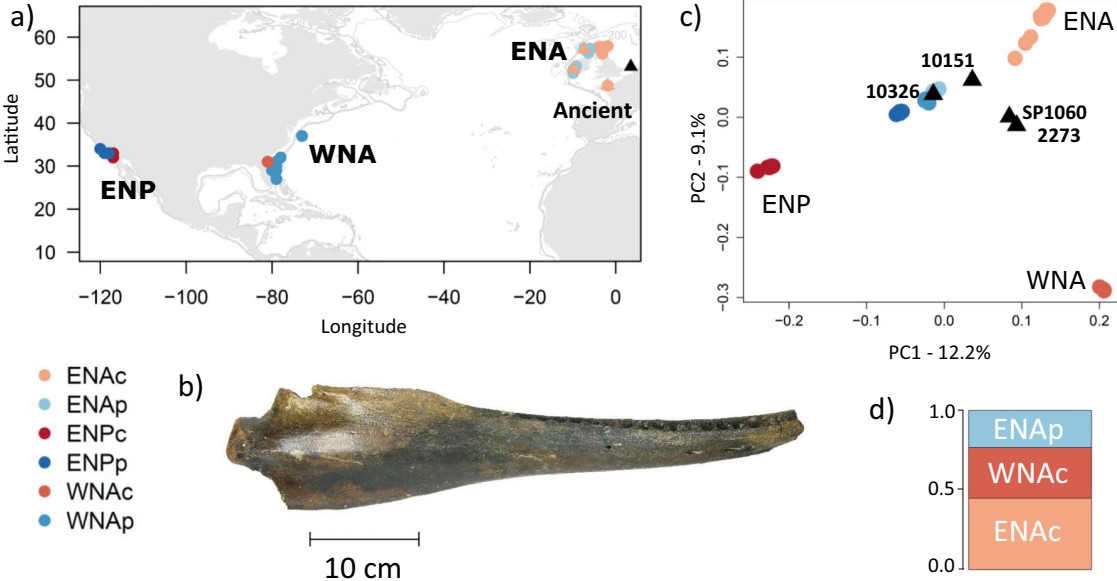

**Fig. 1 | Sampling locations and ancestry of ancient and contemporary bottlenose dolphin individuals. a** Map of sample locations of the four ancient (black triangle) and 60 contemporary coastal (denoted with postscript 'c' and shown in shades of red) and pelagic (postscript 'p' and in shades of blue) bottlenose dolphins in the eastern North Atlantic (ENAc and ENAp), western North Atlantic (WNAc and WNAp) and eastern North Pacific (ENPc and ENPp). Calibrated radiocarbon-dated the age of the ancient samples is as follows in years before present (BP): NMR10326: 8,518-8,346, NMR2273: 7,745-7,572, NMR10151: 7,228-7,036, SP1060: 5,896-5,723; 'NMR' is not indicated on the figure for readability. **b** Mandible of a subfossil

bottlenose dolphin (sample NMR10326) included in this study. **c** Principal component (PC) analysis of pseudo-haploid data from four ancient samples projected on the PCs of 60 contemporary samples, mapped to the killer whale (*Orcinus orca*) reference genome to avoid reference bias and removing transitions, showing first and second PCs based on 624,969 SNPs. The proportion of genetic variance captured by each component is indicated in the axes (see also Figs. S4–S8). **d** Ancestry proportions of sample SP1060 identified using factorial analysis[26]. Source data are provided in the Source Data file.

contemporary individuals and SP1060 at sites with no missing data in SP1060 and both a projection and factorial analysis approaches. Data processing and filtering followed Louis et al.[19] with some modifications specific to ancient DNA data such as recalibration of base quality scores according to DNA damage patterns, which are detailed in the Methods section.

Due to the fragmented and damaged nature of ancient DNA, reads with non-reference alleles are less likely to be mapped than those with reference alleles, creating reference bias. To reduce reference bias, we ran the multivariate analyses with the genomic data mapped to two references genomes: (i) the common bottlenose dolphin reference genome (GenBank: GCA_001922835.1), which is a coastal western North Atlantic (WNA) individual, with relaxed parameters in BWA[28,29] and (ii) the killer whale reference genome (GenBank GCA_000331955.2)[30]. Using relaxed parameters or mapping to another species should help to include a better representation of alternate alleles. We also ran all analyses both including and excluding transitions, as C-T transitions are in excess at molecule ends of the SP1060 sequence reads, reflecting post-mortem deamination damage (see details in the Methods section). Additionally, we down-sampled one contemporary individual from the ENA coastal population to 0.03x to evaluate whether the lower-coverage ancient samples could be pulled towards the middle of the PCAs due to large amounts of missing data. This ENA coastal individual clustered with the other ENA coastal individuals after downsampling in the PC projection (Supplementary Fig. 3a) and single-read sampling approaches (Supplementary Fig. 3b), indicating differences in coverage are not responsible for the observed patterns of variation.

As per Louis et al.[19], the major axis of differentiation is between the Pacific and the Atlantic populations (PC 1, Fig. 1c). Independent genetic drift in each coastal population drives this pattern, while pelagic populations from both oceanic regions cluster in the centre of the PCA. Contemporary coastal samples from other locations in the ENA (northern France, Ireland and West Scotland) form a cline from the pelagic populations to the East Scotland samples. This could be consistent with a northwards range expansion of ENA coastal populations[20]. The second axis of differentiation separated the two Atlantic coastal populations.

Along PC 1, we observe affinity towards the two North Atlantic coastal populations for SP1060 and NMR2273, and to a lesser extent for NMR10151. The ancient samples cluster with the pelagic samples along PC 2 (Fig. 1c). We observe similar results in all analyses, regardless of filtering and mapping strategy (Fig. 1c, Supplementary Figs. 4–8, Supplementary Notes). However, we note some reference bias for the ultra-low coverage samples (Supplementary Figs. 6, 8). The oldest genome (NMR10326, 8610–8243 years BP) clusters with the pelagic populations when mapped to the killer whale reference genome (Fig. 1c), while it does not do so when mapping to the bottlenose dolphin reference genome (Supplementary Figs. 6, 8), likely due to reference bias associated with ancient DNA giving NMR10326 closer affinity on the PCA to the WNA coastal population where the reference genome is from. NMR10326 is however still the closest ancient sample to the pelagic populations (Supplementary Figs. 6, 8). The clustering of genomic data in PCAs reflects shared mean coalescence times and Identity by State[31]. The position of the ancient genomes suggests that the genetic affinity of ancient samples to North Atlantic coastal populations increases as the age of the samples decreases.

Having established that SP1060 is broadly representative of our ancient samples, we focus on SP1060 in the rest of our analyses. We estimated the shared ancestry of SP1060 with contemporary individuals using a factor analysis, which takes genetic drift into account[26]. SP1060 shares the highest ancestry with the ENA coastal population (ENAc, 43%), followed by the WNA coastal (WNAc, 32%) and ENA pelagic (ENAp, 25%) populations (Fig. 1d). Clearly, the inferred ancestry proportions do not represent an admixed ancestry composed of

multiple contemporary populations, as SP1060 was alive at approximately the time of their divergence[32]. Rather, they reflect ancestral genetic variation in SP1060, which later segregated in the different populations[26,32].

The PCA and ancestry results were further confirmed by the sharing of derived alleles identified using $D$-statistic tests[33] of the form $D$(H1,H2; SP1060, *Orca*), with H1 and H2 being two contemporary dolphins. The ancient sample shares a significant excess of derived alleles with both ENA and WNA coastal populations, compared with all other in-group populations (Supplementary Fig. 10a). The value of the statistic $D$(ENAc, WNAc; SP1060, *Orca*) is significantly negative ($Z$-scores of −6.9 to −8.8), indicating that SP1060 is more closely related to the ENA coastal dolphins than to the WNA coastal dolphins. Accordingly, SP1060 shares a higher excess of derived alleles with the ENA coastal population than with the WNA coastal population, in statistics of the form $D$(coastal, pelagic; SP1060, *Orca*) as shown by a stronger non-zero $D$-statistic when the coastal individual is from the ENA than from the WNA.

Having established the broad relationship of ancient samples to contemporary populations, in terms of shared ancestry, we next sought to reconstruct evolutionary history through time as an admixture graph[34]. Testing across all possible histories, we find one admixture graph with no outlier $f$-statistics (i.e. all |Z| were <3, the greatest deviation from 0 was −0.96) using qpBrute (Fig. 2), which explores the space of all possible admixture graphs of a given maximum complexity, under a brute-force approach[35,36]. Graphs were estimated using pseudo-haploid data mapped to the killer whale reference genome and using the killer whale as an outgroup. We find similar topologies when using called genotypes for the contemporary populations only and mapping to the bottlenose dolphin reference genome, indicating that the approach based on pseudo-haploid data is robust (Supplementary Fig. 11).

The best-fitting graph reveals a basal split between the lineage that gave rise to the contemporary North Atlantic coastal populations and SP1060, and the lineage that gave rise to the majority of ancestry in contemporary pelagic populations (Fig. 2). The results indicate that WNA coastal, which has recently been described as its own species, *Tursiops erebennus*[16], and SP1060 are independent lineages. The ENA coastal is depicted as a clade whose ancestry is a mixture of the two ancestral groups leading to SP1060 and WNA coastal. Ancient sample SP1060 is therefore not a direct ancestor of the contemporary ENA coastal dolphins. The ancestry of both North Atlantic contemporary pelagic populations appears to be an admixture of ~30% of the lineage giving rise to the coastal populations, and ~70% from a deeply divergent lineage. Thus, the results provide a useful visualisation of how coastal-associated alleles may have been reintroduced into pelagic populations. The branches leading to the pelagic populations have null or small drift values, consistent with pelagic populations having large ancestral effective population sizes or showing little genetic structure, and indicating minimal drift from a shared ancestral population, consistent with previous demographic inference[19,31,37]. Those inferences showed that pelagic populations in different oceanic regions are closer to each other than to their parapatric coastal populations. They also showed that effective population sizes are larger and more stable through time in pelagic than in coastal populations[19].

## Patterns of selection to coastal habitat in the ancient individual
We have shown that due to its relationship with contemporary populations, the ancient sample SP1060 can provide a unique temporal resolution to the chronology and pace of genetic changes associated with the colonisation of coastal habitat in the eastern North Atlantic. Thus, we investigated and compared patterns of variation in SP1060 and contemporary populations at sites previously identified as evolving under parallel linked selection[19]. Coastal-associated genotypes at sites inferred as evolving under parallel linked selection in coastal

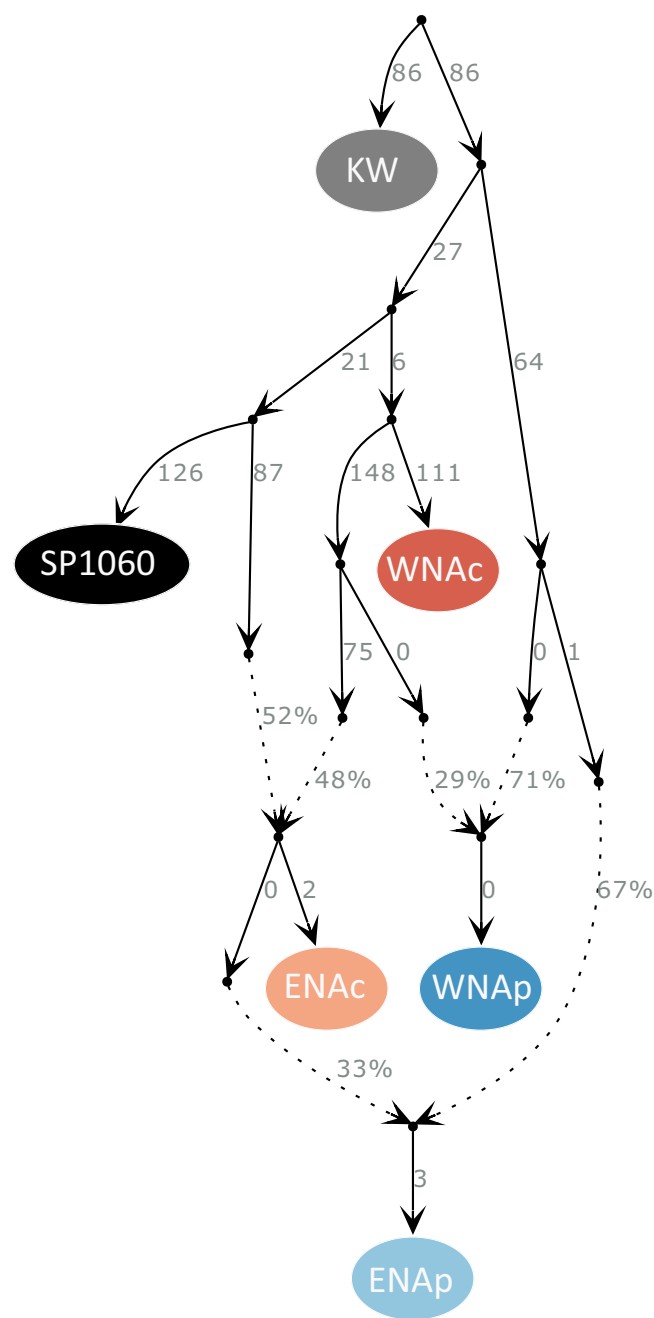

**Fig. 2 | Evolutionary relationships between the ancient individual SP1060 (5,979-5,626 years BP) and the North Atlantic contemporary bottlenose dolphin populations as inferred using qpBrute.** Solid arrows indicate the relationships between populations/samples and the numbers on their right side correspond to the estimated genetic drift represented by the arrow. This graph was the only one of all possible graph combinations presenting no outlier *f*-statistics (i.e. all |Z| were <3). Populations include eastern North Atlantic coastal (ENAc) and pelagic (ENAp) populations, western North Atlantic coastal (WNAc) and pelagic (WNAp) populations, and the outgroup is the killer whale (KW). Note that the drift value for SP1060 is inflated due to being a single and lower-coverage sample. This inflation is because all alleles found in SP1060 are treated as fixed, therefore singletons and rare alleles in SP1060 that were not shared or were rare in this ancestral population are treated as high frequency or fixed alleles, inflating estimates of drift along the branch to SP1060. Dashed lines are admixture edges and the arrows indicate the inferred direction of admixture, with the numbers reflecting the percentage of ancestry deriving from each lineage. The graph reveals a first split between the lineage that gave rise to the contemporary coastal populations and SP1060, and the lineage that gave rise to the majority of ancestry in contemporary pelagic populations. The WNA coastal and SP1060 are shown as independent lineages. The ENA coastal is depicted as a clade whose ancestry is a mixture of the two ancestral groups leading to SP1060 and WNA coastal. Ancient sample SP1060 is therefore not a direct ancestor of the contemporary ENA coastal dolphins. The ancestry of both North Atlantic contemporary pelagic populations appears to be an admixture of ~30% of the lineage giving rise to the coastal populations, and ~70% from a deeply divergent lineage. Source data are provided in the DataSuds repository.

emergence around 7–6000 years BP[21]. The fossil record shows that the North Sea was mostly an Arctic environment during the deglaciation, and that temperate marine mammal species appeared around 8–7000 years BP[22,23]. Palaeogeographic models indicate that the Southern North Sea was progressively submerged between 9000 and 6000 years BP and transitioned from salt marshes to marine waters[21]. This suggests these environmental changes resulted in the selection of standing genetic variation with rapid changes in population allele frequencies in newly founded coastal populations.

Combining contemporary and ancient samples, we also provide insight into the independence of parallel linked selection in coastal bottlenose dolphins from standing genetic variation present in pelagic populations. Selection would be considered independent if it acted upon standing genetic variation in each derived population (scenario i, Fig. 3d)[4], meaning selection would act along branches that are not shared among the two North Atlantic coastal populations and SP1060 in Fig. 2. It would also be independent if adaptive standing genetic variation was shared before independent selection among coastal populations through gene flow (scenario ii). Selection would not be independent if selection acted upon standing genetic variation in a shared ancestral population (scenario iii)[4], meaning along branches shared among the coastal populations[4].

To identify which of these three scenarios best fits our data, we compared patterns of variation (Fig. 3c and Supplementary Figs. 12, 13) and the neighbour-joining tree (Fig. 3b) obtained for our populations at the sites under parallel linked selection with the predictions under each scenario made by Lee and Coop (2019)[4]. For the SNPs under parallel linked selection, the coastal individuals cluster by population, but with inter-individual variation[19]. SP1060 clusters most closely with two individuals of the ENA coastal population (Fig. 3a), but diverges close to the basal node of all the Atlantic coastal populations (Fig. 3b). Such clustering of individuals by population is the pattern expected under scenario i), which describes an independent selection from standing genetic variation in each population, as it will generate only partially shared haplotypes. In contrast, scenarios (ii) and (iii) would have generated shared haplotypes among populations, and individuals from the coastal populations would be more mixed in the tree. While the Lee and Coop approach is based on trees for each region under selection, we based our analysis on different regions of the genome,

populations[19] are also found in the ancient sample SP1060 (Fig. 3a–c and Supplementary Fig. 12). This observation informs us about the speed of local adaptation. We observe covariance in alleles among contemporary coastal individuals within a region, but not between contemporary coastal and pelagic individuals, consistent with drift from the pelagic populations in the coastal contemporary individuals. However, based on their position in the PCA space, SP1060 and the other ancient samples do not share all the drift (and consequent covariance of alleles) experienced by the coastal populations (Fig. 1c). Yet, the genome of SP1060 shares the coastal-associated variation inferred to have evolved under parallel linked selection in the coastal populations[19] (Fig. 3a, b). The ancient genome shows excess heterozygosity as found in coastal individuals[19], despite its lower coverage (Fig. 3c and Supplementary Figs. 12, 13). SP1060 is dated to 5979–5626 years BP, close or shortly after the colonisation time of coastal waters by bottlenose dolphins in the North Sea as indicated by genetic studies[12,20], the subfossil record[22,23] and coastal marine habitat

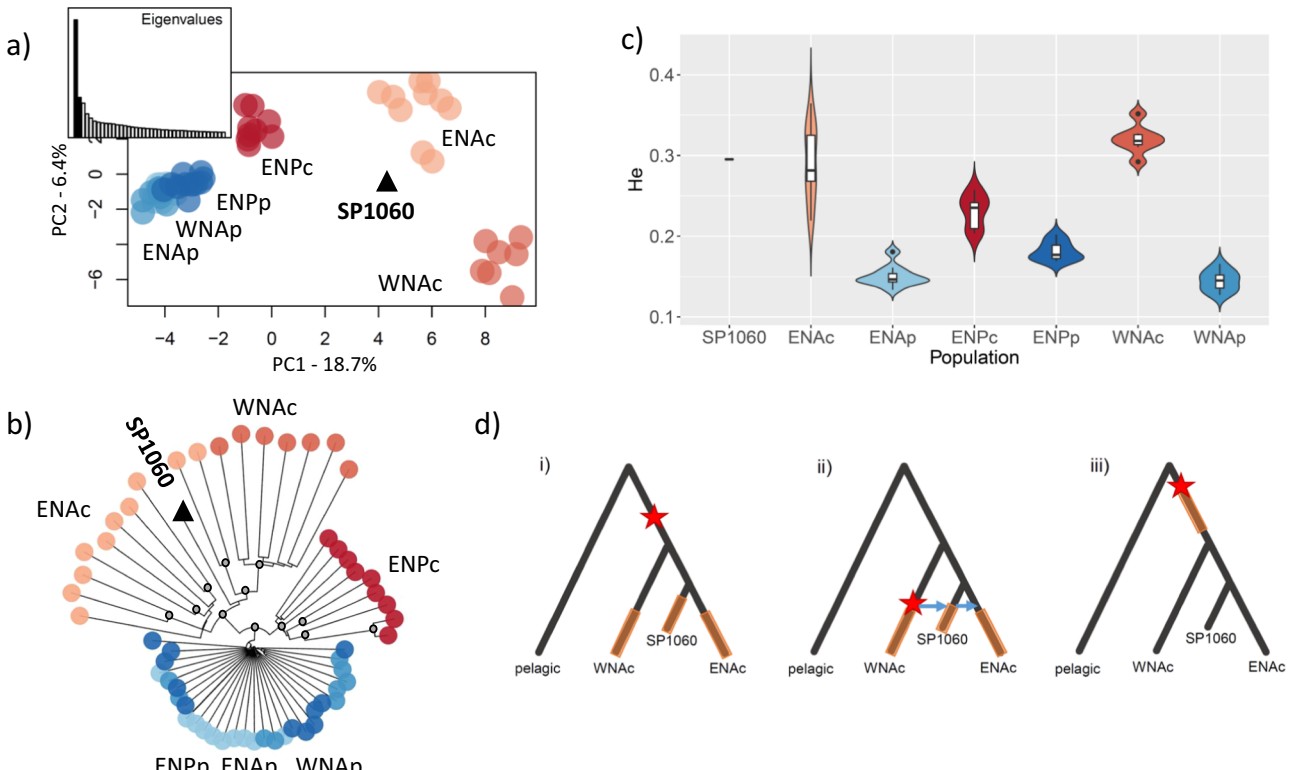

**Fig. 3 | Patterns of genetic variation of the SNPs under repeated selection to coastal habitat in contemporary common bottlenose dolphin individuals and ancient individual SP1060.** These include 2122 SNPs with no missing data in SP1060 out of the 7165 SNPs identified in ref. 19, mapped to the bottlenose dolphin reference genome. Populations include coastal and pelagic ecotypes from the eastern North Atlantic (ENAc and ENAp), western North Atlantic (WNAc and WNAp) and eastern North Pacific (ENPc and ENPp). **a** Principal component analysis including Eigenvalues; the proportion of genetic variance captured by each component is indicated in the axes; and **b** Neighbour-joining tree showing the genetic structure of the common bottlenose dolphin samples for this particular SNP set, with grey circles indicating bootstrap node support values higher than 95%.
**c** Heterozygosity (He) estimated for all the sites under parallel linked selection for each population using angsd 0.921; the violin plots indicate the kernel probability density of the data, the box indicates the interquartile range, and the horizontal marker indicates the median of the data. There is only one data point for SP1060. Heterozygosity is significantly higher in coastal than pelagic populations as shown by two-sided $t$-tests or Wilcoxon tests in the ENP (ENPc $n = 9$, ENPp $n = 11$, $t = 6.54$, df = 12.07, $P = 2.68e{-}05$), ENA (ENAc $n = 13$, ENAp $n = 10$, $W = 130$, $P = 1.75e{-}06$) and

WNA (WNAc $n = 7$, WNAp $n = 10$, $t = 22.23$, df = 10.01, $P = 7.45e{-}10$). Note that genome-wide coastal populations have less diversity than pelagic populations as described in ref. 19. SP1060 shows a mean He of 0.295 which is very close to the mean of 0.291 for ENAc. The same figure with all the data points can be found in Fig. S12b, **d** Illustrations of the three possible scenarios of parallel and non-parallel selection based on ref. 4. The star represents a beneficial mutation and the brown bars the period of time during which selection acted upon standing genetic variation. Scenario (i) highlights standing genetic variation originating in the ancestral population and then being targeted by selection independently in each derived population. Scenario (ii) corresponds to a scenario where standing genetic variation originated in one derived coastal population and was then shared through gene flow with the other coastal populations. Subsequently, selection acted independently in each coastal population to increase the frequency of the adaptive variants. Scenarios (i) and (ii) represent parallel selection. Scenario (iii) shows selection acting upon standing genetic variation in a shared ancestral population. This does not fit parallel selection due to the non-independence of adaptive allele frequencies in the derived populations. Source data are provided in the DataSuds repository and in the Source Data file.

thus assuming that the different regions fit the same selection scenarios. We compared the generated tree with ten trees generated using the same number of randomly sampled neutral SNPs. These latter trees do not show clustering by ecotypes nor strong differences in branch length between coastal and pelagic individuals in contrast to the tree based on the sites under selection (Supplementary Fig. 14). However, we acknowledge that the random sampling of SNPs across the genome may bias towards reconstructing the population consensus tree. Similarly, the topology of the concatenated coastal-associated SNPs may not capture variation in local topologies which are informative of the mode of adaptation. We could not investigate haplotypic differences as in Lee and Coop (2019)[4] due to the lower coverage of the ancient sample, which would impact phasing.

We observe long branches in the neighbour-joining tree for all the coastal individuals (Fig. 3b). We previously identified these regions as representing old variants (0.6 to 2.3 million years); based upon the excess of mutations that had accumulated within them, compared with the genome-wide average[19]. We find coastal-associated variants in the genome of SP1060, despite it not sharing all the drift experienced by

the ENA coastal population after divergence from the pelagic population (Fig. 1c). Thus, we hypothesise there was an abundance of coastal-associated standing genetic variation in the ancestral source population; for example, through a large influx of standing genetic variation from coastal populations in refugia, such as the Mediterranean Sea. Habitat models infer the Mediterranean Sea to have been a suitable habitat for bottlenose dolphins during the LGM[20].

Although paleogenomics has been used to understand patterns of adaptation in humans[38] and domesticated species[39], it has so far mainly been used to understand demographic history in non-model wild species[40–42] and only in one study to investigate the temporal dynamics of parallel evolution[43]. In our study, we harness the power of paleogenomics to disentangle the mode of parallel selection through direct observations of the chronology and genetic changes associated with the formation of coastal ecotype populations of bottlenose dolphins. Using subfossil samples pre-dating much of the drift experienced by coastal populations enables us to differentiate drift from coastal adaptation. Overall, our results suggest that coastal variants represent balanced polymorphisms *sensu* ref. 44 that were rapidly and

repeatedly sieved from standing variation by ecological selection. We thereby provide rare direct evidence of rapid adaptation to newly emerged habitats from standing genetic variation. Our study contributes to the debates of the extent to which parallel evolution is common or rare, and to the role of standing genetic variation in favouring local parallel adaptation[4,45]. We show that paleogenomics can help solve these debates, and our study can be used as a roadmap for future studies of parallel evolution.

## Methods

### Ethics
We confirm our research complies with all relevant ethical regulations and was approved by the animal ethics committee of the School of Biology at the University of St Andrews on 26 July 2018 https://www.st-andrews.ac.uk/research/environment/committees/awerb/. The three new contemporary dolphin samples analysed in this study were collected under the relevant permits of each country. Biopsy sampling was carried out under licence from the National Parks and Wildlife Service in Ireland and under licence from the Direction Regionale de l'Environnement, de l'Amenagement et du Logement in France.

### Samples
**Ancient sample collection and geological analyses.** The four common bottlenose dolphin subfossil samples were dredged from the southern part of the North Sea (i.e. Southern Bight and Smiths Knoll) by commercial trawlers, and were stored at the Natural History Museum of Rotterdam (Supplementary Table 1). We collected bone powder in a sterile environment from the four specimens following the Natural History Museum of Rotterdam guidelines. Radiocarbon dating was performed in previous studies at the Klaus-Tschira-AMS facility for SP1060[46] and at the University of Groningen for NMR2273 and NMR10326[47]. For NMR10151, we performed radiocarbon dating at the University of Oxford. We re-calibrated the age of all the samples using the marine20 correction[48] in CALIB 8.2[49] applying a ΔR (i.e. localised reservoir correction) of $8 \pm 38$ as estimated for odontocete bones in Norway[50].

**Contemporary sample collection.** We used whole-genome re-sequencing data from 57 bottlenose dolphins[19] and generated new data from three coastal individuals from the eastern North Atlantic. Two individuals were sampled using biopsy sampling in Northern France (Normandy) and Ireland (Shannon estuary). The third sample was collected from a stranded animal in West Scotland. Ecotype was identified previously using microsatellite markers[51]. Samples were either frozen or stored in 95% ethanol.

### Laboratory work
#### Ancient DNA labwork
**Ancient sample SP1060.** Laboratory work was conducted in a dedicated ancient DNA facility at the Max Planck Institute for Evolutionary Anthropology in Leipzig for sample SP1060 (Supplementary Table 1). The sample was treated with 0.5% bleach before extraction. Extraction and single-strand library preparation of the ancient sample are detailed in ref. 52. We sequenced the sample on two lanes of HiSeq 2500 with the 80 bp SE technology at the Danish National High-throughput Sequencing Centre of Copenhagen University.

**Ancient samples NMR2273, NMR10326 and NMR10151.** We processed the three additional samples (Supplementary Table 1) at a dedicated ancient DNA facility at the Centre for GeoGenetics, University of Copenhagen. We extracted DNA from around 50 mg of bone powder, twice, both without and with a 1% bleach treatment. We incubated the samples overnight under motion at 55 °C in 500 µl of extraction buffer (0.45 M EDTA, 0.1 M UREA, 150 µg proteinase K). After centrifugation of the samples at $1700 \times g$ for 5 min, we collected the supernatant, concentrated and purified it using a Zymo-Spin V reservoir (Zymo Research Irvine, CA, USA) and Qiagen MinElute spin column (Qiagen, Inc., Valencia, CA, USA).

We built double-stranded libraries for these three samples following the blunt-end single-tube library building (BEST) protocol[53]. We index amplified the libraries with the following thermocycling conditions: 2 min at 95 °C, followed by 20 to 26 cycles of 30 s at 95 °C, 30 s at 60 °C, and 1 min 50 s at 72 °C, and a final, 10-min elongation step at 72 °C. We purified PCR products using Qiagen MinElute spin columns (Qiagen, Hilden, Germany) following the manufacturer's instructions and eluted in 20 µl EB.

We performed two rounds of in-solution enrichment capture on pooled libraries from different PCR reactions to increase complexity using a custom capture-enrichment kit produced by MYbaits (Mybaits Whole-Genome Enrichment, MYcroarray, Ann Arbor)[54]. RNA bait libraries were constructed by transcribing a fragmented contemporary high molecular weight bottlenose dolphin DNA (sample was from ref. 30). We hybridised ancient DNA libraries to these RNA baits in a reaction containing adaptor blockers for 24 h. We used streptavidin-coated magnetic beads to retain hybridised fragments and discard unbound DNA[55] following the manufacturer's instruction (online version 3.02 – July 2016). We amplified captured libraries in two PCR reactions using KAPA HiFi polymerase (Kapa Biosystems) with the following thermocycling conditions: 2 min at 98 °C, followed by 14 cycles of 20 s at 98 °C, 30 s at 60 °C and 20 s at 72 °C, and a final, 5-min elongation step at 72 °C. We performed hybridisation capture a second time on the pool of PCR products. We purified, quantified and pooled the double-capture libraries at equimolarity. We sequenced the libraries on one lane of HiSeq 2500 with the 80 bp SE technology at the Danish National High-throughput Sequencing Centre of Copenhagen University.

**Contemporary sample DNA labwork.** We extracted DNA from epidermal tissue using standard phenol:chloroform extraction method[56]. We then sheared genomic DNA to an average size of ~500 bp using a Diagenode Bioruptor Pico sonication device. We built libraries on the DNA samples following the BEST protocol[53]. We pooled the libraries at equimolarity, together with samples from other projects. We sequenced the pool on an Illumina HiSeq 4000 with the 80 bp SE technology at the Danish National High-throughput Sequencing Centre of Copenhagen University.

### Data processing
**Read trimming.** We used Illumina's CASAVA 1.8.2 software for the four ancient samples and the three new contemporary samples to convert Illumina's *.bcl files to fastq and perform demultiplexing allowing for no mismatch in the 6-nucleotide indices used for barcoding. For the ancient samples, we processed sequencing reads within the generated fastq files with ADAPTER-REMOVAL v.2[57] to trim residual adapter sequence contamination and to remove adapter dimer sequences as well as low-quality stretches at 3′ ends (i.e. consecutive stretches of N's and of bases with a quality score of 2 or lower). We discarded sequence reads that were ≤30 bp following trimming. For the 57 contemporary samples from ref. 19 and the three new contemporary samples generated during this study, we trimmed sequencing reads using Trimmomatic v.0.32[58] using default parameters, and we discarded sequence reads shorter than 75 bp- see details in ref. 19.

**Mitochondrial genomes.** We first mapped the filtered reads to a modified version of the bottlenose dolphin mitochondrial genome from Scotland (GenBank KF570351.1)[59] as per ref. 60. For the contemporary samples, we used BWA (v. 0.7.15) mem and default parameters[61]. For the ancient samples, we used BWA aln with the seed disabled (-l 1024) to include reads with post-mortem damage at the read ends[29]. We also mapped the reads to two additional individuals, to

ensure there was no reference bias, which can be an issue with ancient DNA, i.e. one individual from the Black Sea (GenBank KF570326.1) and one individual from the western North Atlantic (GenBank KF570378.1)[59]. We compared the sequences and checked they were the same. We discarded PCR duplicates using the rmdup function of samtools v. 1.2[62,63] for the ancient samples and using picard-tools v. 2.1.0[64] for the contemporary samples. We constructed a consensus sequence using the doFasta function in ANGSD v. 0.913[25] setting the mapping quality to 25 and the phred score to 30. We changed nucleotide sites to "N" if a single nucleotide did not represent >75% of the reads. We aligned all sequences with ClustalW in MEGA X[65] with 30 sequences from ref. 20 and 48 sequences from ref. 59 and we visually inspected them for indel and reading frames.

**Nuclear genomes.** We extracted reads that did not map to the mitochondrial genome from the bam file and converted them into a fastq file using samtools v. 1.2 and picard-tools v.2.1.0. For the contemporary samples, we processed the data as described in ref. 19. In short, it involved the same steps as described below for the ancient samples, without the tweaks to decrease reference bias. For the ancient samples, we then mapped these reads to the reference bottlenose dolphin genome assembly (GenBank: GCA_001922835.1, NIST Tur_tru v1) using BWA (v. 0.7.15) aln[61] with the seed disabled (-l 1024) and both default and adjusted parameters which were shown to reduce reference bias and include a better representation of alternate alleles[28]. Those parameters were an edit distance (n) of 0.01 and a maximum number of gaps (o) of 2. To further evaluate reference bias, we also mapped the reads of the ancient and contemporary samples to the killer whale genome assembly (GenBank GCA_000331955.2)[30] using default parameters in BWA. Then, we kept only the mapped reads with a mapping quality of at least 25. We removed repeated regions as identified using RepeatMasker[66], regions of excessive coverage and mapping artefacts (sites which were out of Hardy Weinberg Equilibrium with negative inbreeding coefficients in all populations as they could be paralogues or other mapping artefacts), and the sex chromosomes using a combination of bedtools v. 2.25.0 and samtools (see details in ref. 19). We also removed all the scaffolds which were shorter than 10 Mbp.

We inferred endogenous content as the percentage of sequencing reads mapping to the reference genome after base quality filtering of q-score 30 and read duplicate removal (Supplementary Table 2).

## Data analyses

**Assessing post-mortem DNA damage and contamination.** We used MapDamage 2.0[67] to characterise post-mortem DNA damage and thereby check the authenticity of the data. Analyses revealed that the sequencing reads showed post-mortem damage characteristic of ancient DNA (Supplementary Fig. 1) including an excess of C- > T transitions at both termini for SP1060 and an excess of C- > T transitions at the 5' termini due to deamination and G- > A transitions at the 3' termini for the three other samples on which double-stranded libraries were built. Therefore, we used MapDamage to rescale base quality scores according to damage patterns when phred quality score ≥30 both for the mitochondrial and nuclear data. We then refiltered the bam files to keep only bases with a phred quality score ≥30. We ran all downstream analyses both including all sites and only including transversions. Results were the same, unless otherwise specified.

We detected no heterozygous sites in the mitochondrial genomes of the ancient samples, indicating an absence of contamination from present-day DNA.

## Mitochondrial DNA phylogeny

**Time-calibrated models using the mtDNA.** We first aligned mitochondrial genomes of *T. truncatus*, extracted from carbon-dated ancient subfossils (*n* = 4), contemporary samples (*n* = 70; including the 57 samples from ref. 19, three newly sequenced ones plus ten samples

from Nykänen et al. (2019)[20], see supplementary data on the DataSuds repository), or downloaded from GenBank (*n* = 68, see Supplementary Table 3 and Supplementary Data)[59] in software MEGA X[65] with a *T. aduncus* reference mitogenome (KF570335.1) downloaded from GenBank. Then, we built a topology tree in MrBayes v.3.2.7a[68,69] with these sequences, using 2,000,000 Markov Chain Monte Carlo (MCMC) samples and 25% burn-in, after the initial model selection for the best substitution scheme, ref. 70 with a proportion of invariable sites and Gamma distributed rate variation among sites, in jModelTest2[71,72]. The inspection of the resulting consensus tree revealed that one sample (sample 117696 collected from the coastal eastern North Pacific) was very differentiated from the rest of the samples forming its own monophyletic branch. Therefore, we also ran the topology tree without this sample using the same settings as before. We then inspected the consensus trees from the two analyses (with and without 117,696) to find the two most evolutionarily distant *T. truncatus* haplotypes from both trees using the R-function 'cophenetic' from *ape* R package v.5.0[73] in R v4.0.5. These sequences represented a sample obtained from coastal western North Atlantic (sample WNAC8) and sample 117696; and sample WNAC8 and a sample originating from the eastern Mediterranean Sea (sample EMED6).

To estimate time-calibrated phylogenies, we used a two-step methodology by ref. 60. The purpose of the first step (delphinid tree model) was to estimate a calibration point (divergence of the two most evolutionarily distant *T. truncatus* haplotypes) for the second step (*T. truncatus* tree model). The two steps are described below.

**Delphinid phylogeny.** We aligned thirteen protein-coding genes of the mitochondrial genome from the two most divergent *T. truncatus* haplotypes (WNAC8 and 117696, or WNAC8 and EMED6) with the gene sequences of 19 delphinid species downloaded from GenBank (Supplementary Table 3)[59,60,74-78]. We used the 'greedy' search in PartitionFinder (v1.1.0)[79] to find the best partitioning scheme for each gene, and we built two time-calibrated phylogenetic trees (one with and one without the sample 117696) in BEAST2 v.2.6[80] with three different data partitions for nucleotide substitution models (Supplementary Table 4). We used all codon positions and only the third codon positions, both with a single strict clock model based on the results from a previous study[20], and concatenated the individual gene trees into a single phylogeny to find the time to a most recent common ancestor (TMRCA) between the two most divergent *T. truncatus* haplotypes. We ran two independent models for each scenario with 40,000,000 MCMC steps, 10% pre-burn-in, and a sampling frequency of 4000. We used the divergence between Monodontidae and Delphinidae (mean = 10.08 Myr, SD = 1.413[81]) to calibrate the root of the tree and we used a Calibrated Yule prior for the branching rate. We checked the convergence of MCMC chains and the Effective Sample Size (ESS) values relating to the model parameters in Tracer v.1.7.1[82]. After verifying convergence, we used LogCombiner v.2.6 and TreeAnnotator v.2.6[83] to combine and summarise the trees, respectively.

***T. truncatus* phylogeny.** We extracted 13 protein-coding gene regions from the 142 common bottlenose dolphin samples in order to estimate the coalescence times of different *T. truncatus* clades. We removed duplicate haplotypes from this dataset, thus eliminating one ancient sample, NMR2273 (duplicate with another ancient haplotype SP1060), and 51 contemporary samples. We determined the best nucleotide partitioning scheme for the remaining 90 haplotypes using PartitionFinder (v1.1.0)[79] (Supplementary Table 4). We used all codon positions and, as in ref. 60, only the third codon positions of the genes to minimise any possible effect of incomplete purifying selection on the coalescence times of the tree[84,85]. We built time-calibrated phylogenies in BEAST2 v.2.6[80] with two different data partitions for nucleotide substitution models (Supplementary Table 4). We applied a common strict clock model for all of the genes, following ref. 60 and

ref. 20. For each tree built, we performed two independent runs with 100,000,000 MCMC steps, 10% pre-burn-in, and a sampling frequency of 10,000. We used the time to the most recent common ancestor (TMRCA) of all of the haplotypes, derived in the previous step when estimating the delphinid tree, to calibrate the root of the tree and a coalescent prior with constant population size was used for the tree branching rate. In addition, we applied tip calibrations with the carbon-dated subfossil samples. We ran altogether eight different scenarios for the trees, with and without the ancient sample NMR10151 which had low mitogenome coverage (2.3x) with 25% of the bases missing compared to the other subfossils, and with and without the contemporary sample 117696. We describe the different tree scenarios in Supplementary Table 5, and details of the priors can be found in the BEAST2 input.xml files on the DataSuds repository. For all tree models, we inspected the convergence of chains and model performance in Tracer[82] after running each model twice, and we used LogCombiner and TreeAnnotator[83] to combine the log- and tree-files and summarise the trees, respectively. We draw the resulting summary trees using FigTree v1.44 software (http://tree.bio.ed.ac.uk/).

**Population structure.** We used genotype likelihoods or pseudo-haploid data in ANGSD v.0.921[25] to take into account differences in coverage between samples, where possible, or based our analyses on allele frequencies from called genotypes. We generated a vcf file including the contemporary individuals and SP1060 following the filtering steps as in ref. 19 and given in the supplementary scripts. In addition, in the vcf file, we kept only sites with data with 90% of the individuals and sites with no missing data in the ancient sample.

**Projection of the ancient samples on the principal components (PC).** Note that we ran all the analyses mapping to the common bottlenose dolphin reference genome with relaxed BWA parameters and to the killer whale reference genome to check there was no effect of reference bias. Results were the same, unless otherwise specified.

We performed PCA projections of the ancient genomes on the principal components segregating the contemporary genome with smartpca from the eigensoft v.7.2.0 package[24] using the option lsqproject. We ran smartpca both on (i) the contemporary samples and SP1060 and diploid genotype calls, including the contemporary individuals and (ii) SP1060 and the three other ancient samples using pseudo-haploid genotypes generated in ANGSD v.0.921.

We converted the vcf genotype file to the eigenstrat format using the utility vcf2eigenstrat.py from https://github.com/mathii/gdc/blob/master/vcf2eigenstrat.py.

For the pseudo-haploid genotype calls, we first used the following filters in ANGSD, including a MAF of 0.05 (minMinor 6) and data in 25% of the individuals (maxMis16): -dohaplocall 1 -doCounts 1 -minMapQ 25 -minQ 30 -C 50 -baq 1 -minMinor 6 -maxMis 16 -skipTriallelic 1 -remove_bads 1 -uniqueOnly 1 -ref ref.fa.

We then used the utility haplotoPlink from ANGSD to transform the haplo file to Plink tped and tfam. We used Plink v.1.9 to transform those into bed/bim/fam files which we converted into eigenstrat using convertf from eigensoft v.6.1.3. We used the options–allow-extra-chr–chr-set 85 (killer whale reference genome scaffolds >10 Mbp) or 56 (bottlenose dolphin reference genome scaffolds >10 Mbp) in Plink.

**Factorial analysis (tfa).** We also ran the tfa method, which takes drift into account[26] on the diploid genotypes only, including the contemporary individuals and SP1060 and sites with no missing data in SP1060, mapped to the bottlenose dolphin reference genome with relaxed parameters. We imputed missing values for the contemporary samples using the R package LEA v.3.2.0 for the run with the highest likelihood for K = 6 (this value was previously described as the best-fitted number of populations[19]) using the 'impute' function[86]. We also adjusted the genotypic data for coverage, as described in the method

using the function "coverage_adjust" of the tfa R package, which uses a latent factor regression model. We used the function choose_lambda() to decide the lambda value for which the effect of sample age was removed from the fifth factor of the tfa analysis, similar values were obtained when choosing smaller factor values. We ran the tfa for K = 5 as the two pelagic Atlantic populations are very closely related. We also estimated ancestry coefficients for SP1060 including iteratively all populations. SP1060 did not show any ancestry relationship with the Pacific populations. Therefore, we estimated the ancestry coefficients for SP1060 including ENAc, WNAc and ENAp populations (including WNAp instead of ENAp gave the same results, likely due to the two populations being closely related).

**ANGSD single-read sampling PCA approach.** We also ran the single-read PCA method in ANGSD v.0.921, which involves random sampling of a single read for each sample at each site. In this analysis, the ancient samples were included in the PC computations and not projected onto PCs of contemporary samples. This provides a quality control measure, such as if the ancient samples presented sequencing or sequence data processing errors, they would appear as outliers in the PCA in comparison with the contemporary samples. We ran the analyses, with and without the transitions, and on the data mapped to the bottlenose dolphin reference genome and the killer whale reference genome, on (i) the 60 contemporary individuals and SP1060, allowing no missing data, (ii) the 60 contemporary individuals, SP1060 and NMR10151 allowing no missing data, (iii) the 60 contemporary individuals, SP1060, NMR2273 and NMR10326 allowing for missing data in one individual and (iv) the 60 contemporary individuals and all four ancient individuals allowing for missing data in two individuals. As the results were consistent, we only present those including all the samples.

We used the following options in ANGSD: samples.bamlist -nThreads 9 -doIBS 1 -doCounts 1 -doMajorMinor 1 -minFreq 0.05 -minInd 61 -maxMis 2 -rmTrans 1 -output01 0 -makeMatrix 1 -doCov 1 -minMapQ 25 -minQ 30 -out out_rmTrans -GL 1 -skipTriallelic 1 -remove_bads 1 -uniqueOnly 1.

**Evolutionary relationships**
**D-statistics analysis.** We used D-statistics to assess the relationships of ancient sample SP1060 to contemporary populations. We first performed D-statistics on all 15 possible combinations including two contemporary samples and the ancient sample, with the killer whale, *Orcinus orca*, as the outgroup. The D-statistic describes an excess of shared derived alleles between taxa which could be the result of introgression or ancestral population structure. It thus allows to detect departure from the 'tree-ness' of a given topology[33,34,87]. We considered that H1 and H2 are two contemporary dolphin populations. We used the D-statistics to evaluate if the data are consistent with the null hypothesis that the tree (((H1,H2), SP1060), *Orca*) is correct and that there has been no gene flow between the ancient sample and neither H1 nor H2. The definition of the D-statistics used here is the one of ref. 33.

$$D = (nABBA - nBABA)/(nABBA + nBABA)$$

where nABBA is the number of sites where only H2 and H3 share a derived allele (ABBA sites) and nBABA is the number of sites where only H1 and H3 share a derived allele (BABA sites). Under the null hypothesis that the given topology is the true topology, we expect an equal number of ABBA and BABA sites and thus D = 0. A statistic differing significantly from 0 indicates either gene flow between one population within the in-group and H3, or that the tree is incorrect. We implemented the tests in ANGSD v.0.921, and sampled a single base at each position of the genome to remove bias caused by differences in sequencing depth and only considering sites covered

in all individuals. We calculated *D*-statistics using one individual per population and we repeated the test three times using a different individual each time. We ran the analyses on the data mapped to the common bottlenose dolphin reference genome with relaxed BWA parameters and to the killer whale reference genome to check for reference bias, and both without and keeping the transitions, to make sure there was no impact from DNA damage patterns. We used the same filters as described above for the PCAs. We assessed the significance of the deviation from 0 using a *Z*-score based on blocked jackknife estimates of the standard deviation of the *D*-statistics (block size was 5 Mb which should be higher than the LD in the populations). This *Z*-score relies on the assumption that the *D*-statistics, under the null hypothesis, is normally distributed with mean 0 and a standard deviation equal to a standard deviation estimate computed using the "delete-m jackknife for unequal m" method described in Busing et al.[88]. An example of a command is: angsd -out ${1}_XchrmHWE_minIndKWcov -doAbbababa 1 -rmTrans 1 -blockSize 5000000 -enhance 1 -bam ${1}_XchrmHWE.filelist -doCounts 1 -useLast 1 -minMapQ 25 -minQ 30 -minInd 4 -maxMis 0 -remove_bads 1 -uniqueOnly 1.

**Admixture graph analysis.** We next reconstructed relationships between SP1060 and the Atlantic contemporary populations (*n* = 37) using an admixture graph analysis[34]. We excluded the Pacific populations from this analysis as there are many other populations in between the regions. We only included the already published ENAc individuals (*n* = 10) and did not include the newly generated three individuals as they belong to different populations[51].

We used a heuristic search algorithm, qpBrute (https://github.com/ekirving/qpbrute)[35,36] to explore the space of all possible admixture graphs fitted to the four Atlantic populations using qpGraph. We built the admixture graph using two datasets: (i) pseudo-haploid data including SP1060 and the contemporary Atlantic populations mapped to the killer whale reference as it minimises reference bias, and (ii) called genotypes for the contemporary Atlantic populations mapped to the bottlenose dolphin reference. We used these two datasets to check results were similar, irrespective of the data type (pseudo-haploid and called genotypes) and the reference genome.

For the pseudo-haploid data, we used 580,589 SNPs and a killer whale genome as the outgroup. We kept the transitions as evolutionary relationships were similar with and without the transitions (Supplementary Fig. 9). We used the following command in ANGSD: angsd -b SP1060_Atlantic_KW_10MbpKW.bamlist -dohaplocall 1 -doCounts 1 -minMapQ 25 -minQ 30 -C 50 -baq 1 -minMinor 4 -maxMis 9 -skipTriallelic 1 -remove_bads 1 -uniqueOnly 1 -ref /path/unplaced.scaf.fa -out Atlantic_KW_SP1060_mapKW.

We then converted the data to the eigenstrat format as described above for smartpca.

For the contemporary called genotype data, we used one Indo-Pacific bottlenose dolphin, *Tursiops aduncus*, individual to root the graph. We filtered the dataset as follows: a mapping quality of 30, a genotype quality of 20, genotype depth of at least 3x, less than 25% of missing data overall, genotype data in at least five individuals in each population, a MAF of 0.05, bi-allelic SNPs only, no missing data in the *T. aduncus* individual constituting the outgroup, scaffolds of at least 10 Mbp, and a distance of at least 5 Kb between SNPs. We obtained 213,488 SNPs. We also set the missing data to less than 10% and got similar results.

We ran qpBrute using the following parameters: outpop: NULL, useallsnps: YES, blgsize: 0.05 (5 Mb which is the block size for Jackniffe), forcezmode: YES, lsqmode: YES, diag: .0001, bigiter: 6, hires: YES, lambdascale: 1, inbreed: YES (for the pseudo-haploid data only).

As we have only one individual to root the graph, we did not attempt to use the allele frequency of the outgroup to normalise the

weighting of each SNP in the ingroup and we used the option output: NULL that is SNPs are flat-weighted. We used the option inbreed:YES for the pseudo-haploid data. This allows for each pseudo-haploid to contribute to only one haplotype when applying the low sample size correction for the *f2*-statistics. This does not work when including only a single pseudo-haploid sample as we did with SP1060, but it should not be an issue as using the default option inbreed:NO gave similar results.

In qpBrute, leaf nodes were added to the graph using a stepwise addition order algorithm. At each step, the insertion of a new node was tested at all branches of the graph, apart from the outgroup branch. All possible admixture combinations were tried where a node could not be added without producing *f4*-statistics outliers (i.e. |Z| ≥3). The sub.graph was discarded when a node could not be inserted via these approaches. Where a node was successfully added, then the remaining nodes were recursively inserted into the graph. For the pseudo-haploid dataset, the package tried all possible 120 starting graph orders. We found only one graph with no *f4* outliers among a total of 3663 unique graphs. For the called genotypes, the package tries all possible 24 starting graph orders and fitted 234 unique admixture graphs to our dataset. We found three possible graphs with no *f4* outliers left, although two of them were mirror graphs. We computed the mean log-likelihoods of the three models and their Bayes Factors using the MCMC algorithm implemented in the R package ADMIXTUREGRAPH v.1.0.2[89] using the default chain settings implemented in qpBrute (two chains, each with two million iterations, five heated chains, a burn-in of 50%, and no thinning). We assessed the convergence of the chains using the output from the R package CODA v.0.19-4[90], also generated in qpBrute. The Bayes factors showed non-significant support of the first model in comparison to the other "mirror" two, which have similar log-likelihoods (Bayes factor of 0.88). We increased the chains to four million iterations with a burn-in of three millions, but it did not help in discriminating between the three graphs.

### Inferences on the SNPs under parallel linked selection

**Patterns of variation.** We investigated and compared patterns of variation in SP1060 and contemporary populations at sites previously identified as evolving under parallel linked selection[19]. We plotted a PCA for the 2122 SNPs under parallel linked selection using the packages adegenet v.2.1.5 (glPCA function) and scales v.1.2.1[91] and a neighbour-joining tree using the R package ape[73]. We performed 100 bootstraps of the tree using the boot.phylo option of the package ape v.5.6.2 and indicated for which nodes bootstrap support values were higher than 95%. We compared the tree for the SNPs under parallel linked selection with ten trees generated using the same number of randomly sampled neutral SNPs, also computing bootstrap support values as described earlier. We also plotted the raw genotypes of the 2122 SNPs using the R packages vcfR v.1.12.0[92] and adegenet[93].

**Heterozygosity estimation.** We estimated heterozygosity for the sites under parallel linked selection to coastal habitat by computing individual site-frequency-spectrum using ANGSD v.0.921, both with and without transitions. The first command was as follows: angsd -i file.bam -anc ancestral_state.fa -dosaf 1 -P 6 -gl 1 -out file -minMapQ 30 -minQ 23 -remove_bads 1 -uniqueOnly 1 We then added '-noTrans 1' to remove transitions. Then we ran: realSFS filet.saf.idx >est.ml.

The heterozygosity is the product of the number of variants in the second column of the est.ml file, which corresponds to the heterozygotes, divided by the total number of sites. We tested whether mean heterozygosity was different between coastal and pelagic populations within a region using two-sided *t*-tests or Wilcoxon tests depending on whether the data satisfied normality and homogeneity of variances.

## Reporting summary

Further information on research design is available in the Nature Portfolio Reporting Summary linked to this article.

## Data availability

Raw sequencing data generated as part of this project are part of Bioproject PRJNA955223. Biosample and SRA accession numbers are provided in Supplementary Table 1 and Supplementary data file 1. Mitochondrial genome sequences are accessible on GenBank at accession numbers OR120165-OR120228 as detailed in Supplementary Data File 1. Supplementary Data file 1 also indicates where the samples are housed and who should be contacted to access the material. We also used the whole-genome re-sequencing data from Bioproject PRJNA724031. We have also included mitogenome sequences from Genbank as indicated in Supplementary Table 3. Source data are provided with this paper in the Source Data file and the data, codes and related documentations that support the findings of this study are openly available in the DataSuds repository (IRD, France) at https://doi.org/10.23708/DABAWD. Data reuse is granted under a CC-BY licence. Source data are provided with this paper.

## Code availability

The data, codes and related documentations that support the findings of this study are openly available in the DataSuds repository (IRD, France) at https://doi.org/10.23708/DABAWD. Data reuse is granted under a CC-BY licence.

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

## Acknowledgements

We especially thank Matthias Meyer for providing the SP1060 library, which allowed the generation of the ancient genome that was central to this study. We thank Ramon Fallon, Joseph Ward and Peter Thorpe from the St Andrews Bioinformatics Unit for help with bioinformatics. Bioinformatics and computational analyses were supported by the University of St Andrews Bioinformatics Unit which is funded by a Wellcome Trust ISSF award [grant 105621/Z/14/Z] and ran on cluster marvin and Crop Diversity HPC. We thank Kelly Roberston for the laboratory work for the contemporary samples from the USA and for arranging shipment. We thank M. Thomas P. Gilbert for our use of the ancient DNA facilities in Copenhagen. We thank Jazmín Ramos Madrigal for advice on data format conversion. We thank everyone involved in sample collection including Conor Ryan for sampling some of the stranded individuals in Ireland, the stranding networks in Ireland and Scotland and the Groupe d'Etude des Cétacés du Cotentin, and the NOAA Southeast and Southwest Fisheries Science Centre field crews and Simon Ingram for biopsy sample collection. Funding for ancient DNA labwork and sequencing was provided by the Total Foundation awarded to M.L. Funding for sample collection in Ireland was provided by Science Foundation Ireland. Funding for labwork and visits to the University of Copenhagen was provided by the Systematics Research Fund, a Godfrey Hewitt mobility award from the European Society for Evolutionary Biology (ESEB), Lerner-Gray Grants for Marine Research. M.L. was supported by a Fyssen Fellowship, Total Foundation, the University of St Andrews and the Greenland Research Council. Contemporary sample DNA extractions were supported by People's Trust for Endangered Species and the sequencing costs were supported by the European Union's Horizon 2020 research and innovation programme under the Marie Skłodowska-Curie grant agreement No. 663830 awarded to A.D.F., by the Total Foundation awarded to M.L., the University of Groningen awarded to M.C.F., and the Marine Alliance for Science and Technology for Scotland and The Russell Trust awarded to O.E.G. M.N. was funded by MASTS and the Crawford Hayes fund. A.D.F. was funded by Marie Skłodowska-Curie grant agreement No. 663830 and the European Research Council grant agreement No. ERC-COG-101045346.

## Author contributions

M.L., M.N. and A.D.F. conceived the study; F.A., S.B., A.B., E.R., J.O.B., K.P., P.E.R., H.v.d.E. collected or curated samples/specimens, M.L., P.K., M.N., M.-H.S.S., N.W. and A.D.F. ran the laboratory work; M.-H.S.S. and NW provided guidance in the ancient lab; M.L., M.N. and F.R. analysed the data; all authors interpreted the data; M.C.F., O.E.G. and A.D.F. supervised the work, M.L. and A.D.F. wrote the manuscript, M.N., P.K., F.A., S.B., A.B., E.D.L., J.O.B., K.P., F.R., E.R., P.E.R., M.-H.S.S., H.v.d.E., N.W., M.C.F. and OE.G. commented and proof-read the manuscript. All authors read and approved the final manuscript.

## Funding

## Competing interests

The authors declare no competing interests.

## Additional information

[1]Centre for Biological Diversity, Sir Harold Mitchell Building and Dyers Brae, University of St Andrews, St Andrews, KY16 9TH Scotland, UK. [2]Globe Institute, University of Copenhagen, Øster Voldgade 5-7, 1350 Copenhagen K, Denmark. [3]Groningen Institute for Evolutionary Life Sciences (GELIFES), University of Groningen, PO Box 11103 CC Groningen, The Netherlands. [4]Greenland Institute of Natural Resources, Kivioq 2, Nuuk 3900, Greenland. [5]Max Planck Institute for Evolutionary Anthropology, Deutscher Platz 6, 04103 Leipzig, Germany. [6]Wellcome Sanger Institute, Wellcome Genome Campus, Hinxton, Cambridge CB10 1SA, UK. [7]Department of Environmental and Biological Sciences, PO Box 111, FI-80101 Joensuu, Finland. [8]School of Biological, Earth and Environmental Sciences, University College Cork, North Mall, Cork, Ireland. [9]Marine Mammal and Turtle Division, Southwest Fisheries Science Center, NOAA, 8901 La Jolla Shores Drive, La Jolla, CA 92037, USA. [10]Irish Whale and Dolphin Group, Kilrush, Co Clare, Ireland. [11]Marine and Freshwater Research Centre, Department of Natural Sciences, School of Science and Computing, Atlantic Technological University, Dublin Road, H91 T8NW Galway, Ireland. [12]Scottish Marine Animal Stranding Scheme, Institute of Biodiversity, Animal Health & Comparative Medicine College of Medical, Veterinary & Life Sciences, University of Glasgow, Glasgow, UK. [13]Natural History Museum Rotterdam, Westzeedijk 345, 3015 AA Rotterdam, Netherlands. [14]Marine Mammal and Turtle Division, Southeast Fisheries Science Center, NOAA, 646 Cajundome Boulevard, Lafayette, LA 70506, USA. [15]Department of Biology, University of Copenhagen, Ole Maaløes Vej 5, 2200 Copenhagen, Denmark. [16]University of York, BioArCh, Environment Building, Wentworth Way, Heslington, York YO10 5DD, UK. [17]MIVEGEC (Université de Montpellier, CNRS 5290, IRD 229) Institut de Recherche pour le Développement (IRD), F-34394 Montpellier, France. [18]Department of Natural History, Norwegian University of Science and Technology (NTNU), NO-7491 Trondheim, Norway. [19]Centre for Ecological and Evolutionary Synthesis (CEES), Department of Biosciences, University of Oslo, 0316 Oslo, Norway. [20]These authors contributed equally: Marie Louis, Petra Korlević, Milaja Nykänen. [21]These authors jointly supervised this work: Michael C. Fontaine, Oscar E. Gaggiotti, Andrew D. Foote. ✉e-mail: marielouis17@hotmail.com; andrew.foote@ntnu.no

