## [Peer Review File · Nature Communications]

Ancient dolphin genomes reveal rapid repeated adaptation to coastal watersReviewers' Comments:

Reviewer #1:

Remarks to the Author:

This manuscript represents an exciting advance on the study of recent adaptation to coastal environments in bottlenose dolphins. By sequencing 4 ancient genomes (3 at ultra-low depth) the authors attempt to disentangle the evolutionary history of coastal and pelagic ecotypes. One of the main results is that one ancient genome (from around the time when or soon after the coastal habitat emerged) harboured many of the putatively coastal adaptive genotypes identified in Louis et al 2021 Science Advances study, potentially indicating that selection occurred rapidly on standing genetic variation. The manuscript is generally well-written and novel, given that paleogenomic studies are still very rare and allow for directly testing evolutionary history with ancestral individuals.

It feels like this manuscript in its current form is more suitable for a short communication or a report than for a full original article. While obtaining four ancient genomes (although most analyses include one, which is common practice in ancient genomics) is undoubtedly an incredible feat in itself, the study does not include many analyses and it's around 3000 words in length. However, it is for the editors to decide whether this is a suitable format for this journal.

I have a two main comments:

- 1) The presentation could be much more exciting, and it lacks context. I think the introduction must mention any known phenotypic/behavioural/environmental differences between the coastal/pelagic ecotypes. A reader would have to refer to the authors' 2021 Science Advances article to get any rationale for why this system is interesting and what is known about it. There is surprisingly little discussion about other systems where ancient genomics has been applied to these kind of questions or directions for future studies.
- 2) While I'm not an expert in ancient genomics, I believe there are some claims in the manuscript that require more caution or at least better discussion so that the average reader can understand the evidence and believe those claims. These are highlighted in the specific comments below with **.

Specific comments

L55 please mention how many million years ago the mid Holocene was

** L56 I don't think this claim is clearly discussed or shown in the main text ('We find a progressive shift from a genetic affinity with pelagic populations towards coastal populations through time'). There are four ancient sequences in this study (three of which are at ultra low coverage). Is this claim in the abstract solely based on the position of the four ancient samples on the PCA? The age of the individuals is not indicated (unless is the shading, but that's not explained in the legend). It is mentioned in the supplementary L511 when describing the single-read PCA approach, but there seems to be no consensus with the other approaches.

L59 It is important to disclose that the youngest ancient genome was the only one assessed for presence of coastal-associated genotypes, otherwise it could seem to support the claim above about a shift towards 'coastal' in the four ancient genomes.

L58 before present should be written in full the first time used in each section, then BP (note L93 also)

L90 I think it is very important to define what the authors mean by parallel early in the manuscript, and explain why they chose to refer to linked selection rather than parallel adaptation or evolution.

L94-L95 these two sentences are key to understanding the rationale of the manuscript and I couldn't understand them. Are the ecotype pairs referring to small populations vs large/widely distributed populations referred to in the second sentence? Otherwise, the 'contrast' between the two sentences

isn't clear.

L97 'ancient ancestry tracts' could be briefly explained for clarity

L104 what's a subfossil? And avoid acronyms like ENA when first used (L104), initially thought this referred to the database! L163 WNA also used for the first time in text.

Fig. 1 – could different symbols be used for ancient vs modern in a and b? Additionally, not sure this is a colour friendly palette. I find it particularly difficult to differentiate between the shades of orange ENP/WNA. Perhaps playing with empty vs filled symbols would help.

L153 I appreciate the explanations for the different methods used and they seem appropriate.

L162 what analyses is 'those analyses' referring to?

L173 why is an ancient sample not included in figure s3 PCA? That would be stronger evidence for the lower sequencing depth not pulling samples towards the middle/ towards the ancient genomes.

L174 I feel like since the authors went through the effort of doing all these analyses the results of the single read sampling PCAs should be shown in the Supplementary. Isn't this shown in Fig. s8?

L185 describing patterns along PC2 before PC1 is not common practice, as PC1 explains more variation.

L237 More basic demographic reconstruction of the modern populations used in this study might help interpreting the admixture graph (PSMC, ROH...). I understand that this has been done before and that's why it is cited, but it could be included here as well for clarity.

** L257 The claim that the ancient samples are less genetically diverged from pelagic populations (based on the pca of fig 1c) seems tenuous. SP1060 clusters with eastern north Atlantic coastal populations on PC1, and in the many other PCAs presented in the supplementary they tend to be in the middle. The ancient samples might cluster even more clearly with coastal populations if the highly divergent coastal/pelagic pops from the Pacific were removed.

L261 Please remind the reader here of the dating of the ancient genome (5,979-5,626 BP).

** L259-263 I'm not sure there is a strong argument for 'rapid selection' here. The introduction mentions that the split time of pelagic and coastal ecotypes is 47,800-4,300 years BP (but this was based on few microsats/mitochondrial snps from Louis et al 2014, so perhaps inaccurate), that the coastal habitat in the North Sea emerged 12-6k BP while the ancient genome is dated at 5.8k, so it is on the very recent end of the split time/ after the habitat emerged. The ancient genome is more 'coastal type' than pelagic (in every metric that was looked at: PCA, admixture props, D statistics), is it possible that the split happened much earlier, and that this ancient genome represents an early, already coastal-adapted individual? It would help to be reminded of the dates and supporting evidence in this paragraph.

L285 I don't think randomly sampling neutral snps and making trees rules out variation in evolutionary history across different putatively coastal-adaptation genomic regions. The current method only shows the 'average' pattern across all outlier snps/genomic regions. In contrast, Lee & Coop style methods make inferences about each region and quantify how many fit each scenario (independent, standing variation, or gene flow origin). This should at least be discussed, especially given that there is clearly gene admixture between modern populations. Additionally, I think adding a panel to Fig. 3 with Fig. S12 could be of value.

** L296 what is the support for the claim that there is 'low drift from pelagic population' come from?

In the legend of Fig. 2 it says that the drift value of the ancient genome is inflated due to being a single sample and of lower coverage, so how can we know an accurate drift estimate.

L299 Perhaps coastal population refugia should be mentioned in the introduction

L305 'we have disentangled neutral and selective processes' is quite vague and then only a selective process is mentioned in the next sentence, best specify.

Reviewer #2:

Remarks to the Author:

Ancient dolphin genomes reveal rapid repeated adaptation to coastal waters

Very interesting and well written manuscript investigating the independent repeated adaptation of bottlenose dolphins to coastal habitats using contemporary and subfossil genomes.

What are the noteworthy results?

The authors used a combination of 60 contemporary and 4 subfossil genomes to provide insights on the chronology and independent parallel selection in the formation of coastal bottlenose dolphins. The authors conclude that coastal bottlenose dolphin populations have evolved rapidly and independently through parallel selection from standing genetic variation found in ancestral pelagic populations, thus providing evidence of rapid adaptation to newly emerged habitat.

Will the work be of significance to the field and related fields? How does it compare to the established literature? If the work is not original, please provide relevant references.

Although the divergence of coastal bottlenose dolphins from pelagic populations has been investigated in previous literature, this article is the first one to use subfossil samples to explore the genetic changes associated with the formation of coastal bottlenose dolphins, providing unique genomic data and robust evidence of parallel independent selection. These results are of great importance to our understanding of bottlenose dolphin population dynamics and they represent a significant contribution in evolutionary biology by providing rare evidence of rapid adaptation to new habitats from standing genetic variation.

Does the work support the conclusions and claims, or is additional evidence needed?

I think the data and analysis supports very well the conclusions presented in the manuscript. One of the highlights and what makes in my opinion the manuscript robust, is that the authors used subfossil genomes from a period when bottlenose dolphins were starting to colonize new coastal habitats following post-glacial sea-rise, providing strong evidence for the claims proposed in the article.

Are there any flaws in the data analysis, interpretation and conclusions? Do these prohibit publication or require revision?

Is the methodology sound? Does the work meet the expected standards in your field?

I did not find any flaws in the data analysis, interpretation or conclusions. However, I have not worked using ancient samples before and I have not performed many of the analyses used in the manuscript. Nevertheless, from my point of view the data analysis is very rigorous and scientifically sound. I believe the manuscript meets the standards and would be a great contribution to evolutionary biology in general.

Is there enough detail provided in the methods for the work to be reproduced?

Yes, the authors provide a comprehensive section in materials and methods and all the supplementary data needed to access the genomes used in the study.

Reviewer #3:

Remarks to the Author:

Dear authors,

In my opinion your paper represent an important case study on how to incorporate paleogenomic data to address the origin and evolution of populations within a marine mammal, one that appears to have convergent evolution at the population level. The results are quite interesting, and tie together your detailed analyses with potentially related ecological changes linked to sea level rise since the last glacial maximum. That said, the results themselves will probably be of main interest to expert that work on delphinids (and maybe other cetaceans), whereas the methods used to come to your conclusions may be of broader interest.

The present study appears to differ from previous work from the lead and other authors in its incorporation of subfossil data as well as using methods and models that allow for different hypotheses to be tested that account for apparent convergent molecular evolution in coastal populations of *Tursiops truncatus*.

My main criticism of your approach is the change in sampling of individuals in the analyses that infer gene flow/divergence between present and past populations of *T. truncatus*. In the Bayesian tree (S2) extant pelagic and coastal populations are intermingled with Black Sea and Mediterranean individuals/populations (although I am unsure if any of these represent coastal populations). Thus it would seems appropriate to include these populations in subsequent analyses. I am guessing that their exclusion is related to the assumption? that the divergence of coastal populations represented in the Northeast Atlantic occurred in that region during the past 20K years, and that the Mediterranean and Black Sea Populations are peripheral to this question. If so, that basis for sampling in the different analyses could be better explained.

On a related note, I think your paper could be improved if you better focused on the unique insights provided by the subfossil genomic data. Could the inclusion of extant Mediterranean dolphins (in place of the subfossil SP1060) have yielded essentially identical results? I found it noteworthy that figure S11 (which excluded fossil data) is very similar to fig. 2 (which does). In general, I think your paper could be improved if it showed how a statistical model of fit (or difference between competing evolutionary trees of populations) becomes stronger/more clear cut with the addition of the fossil data. No doubt the acquisition and sequencing of subfossil data required substantial effort on your part, and clearly showing the impact of such data might help encourage other teams to collect such data for their clade/population of interest.

Other Comments

1) Largely citing previous work, the divergence of the Northeast Atlantic coastal populations of *T. truncatus* is correlated with the rise in sea level after the last glacial maximum and the flooding of Doggerland by the North Sea. While this is certainly possible, there are always coastal habitats regardless of the position of sea level. What changes is the complexity and type of coastal habitats (e.g. estuaries are associated with flooding of river valleys) and the total area of shallow marine environments (depending on the position of sea level relative to the shelf break). A few more sentences explaining why this rise in sea level is expected/predicted to have caused a change in selection would be helpful.

2) in Fig. S11, a, b, and c are not explained. Is one of these hypotheses more favored than the other?

Reviewer #4:

Remarks to the Author:

This is a nice straightforward analysis providing an interesting empirical look at parallel evolution in coastal dolphin populations. The methodology is quite robust and the findings are well supported. I have a few minor comments to improve clarity and statistical robustness. It is fundamentally appropriate for Nat Comms with a few edits. Thanks to the authors for an interesting read.

211–213: This is a bit confusing, the text implies comparison between ENA and WNA coastal populations regarding the ancient sample, but the d-stat form indicates pelagic samples. Is the finding that with ENA vs WNA in the “coastal” position, ENA has the stronger non-zero d-statistic? Is that with the same pelagic sample(s)? That seems to be the case but could be clarified.

Figure 2 – it would be useful to provide at least one z-score to demonstrate the quality of the fit. At the least, please define “outlier f-statistics” in the main text (presumably $|Z| > 3$ as in the supplement). Also this figure is a bit sparse. I would consider annotating it further with some of the description from the main text interpretations.

248: Can you please explicate further and/or provide a citation for why lower coverage and a single sample would artifactually inflate the inferred drift parameters?

257: Figure 1c isn't totally clear on the fact that SP1060 (the one actually included in this analysis) is substantially less diverged from the pelagic populations compared with ENAc. I would generally prefer that position on a PCA be used as a hypothesis and then tested with explicit statistical methods. Figure 1d seems to indicate the pattern more clearly. Additionally, a quick f_4 statistic in the form $f_4((SP1060, ENAc), pelagic, OG)$ should be clarifying as to whether SP1060 is more pelagic-like than is ENAc.

260: Heterozygosity needs to be more robustly established to make this statement. With a low-coverage sample, depending on the specific strategy used, sequencing error could more easily affect called genotypes than in high-coverage modern samples. In fact, there is no information that I can find in the supplement about the generation of the called genotypes, do they derive from the Louis et al 2021 methodology? This is very important to include. The heterozygosity estimation also needs to be done very carefully in light of potential coverage-based artifacts linked to sequencing error. The visual of S12 as cited is not sufficient to quantitatively demonstrate higher heterozygosity in the ancient sample.

268: This sentence is quite confusing, although the meaning becomes clear through the next 2 scenarios. The most useful approach would be to illustrate these three scenarios schematically.

276: The visual in fig S12 is very difficult to interpret in light of the three scenarios presented. It's possible to eyeball it and detect some kind of clustering, but this isn't a terribly robust approach.

280–281: This contradicts 3b, which if interpreted in a hierarchical clustering or phylogenetic sense, shows that SP1060 is basal to WNAC with ENAc as the more divergent group. Please clarify, and again, statistical assessment of affinity to groups would be preferable to interpreting the visual in 3a.

306: “balanced” slightly implies balancing selection, which is not in evidence. Consider rephrasing.

REVIEWER COMMENTS

Reviewer #1 (Remarks to the Author):

This manuscript represents an exciting advance on the study of recent adaptation to coastal environments in bottlenose dolphins. By sequencing 4 ancient genomes (3 at ultra-low depth) the authors attempt to disentangle the evolutionary history of coastal and pelagic ecotypes. One of the main results is that one ancient genome (from around the time when or soon after the coastal habitat emerged) harboured many of the putatively coastal adaptive genotypes identified in Louis et al 2021 Science Advances study, potentially indicating that selection occurred rapidly on standing genetic variation. The manuscript is generally well-written and novel, given that pelegenomic studies are still very rare and allow for directly testing evolutionary history with ancestral individuals.

It feels like this manuscript in its current form is more suitable for a short communication or a report than for a full original article. While obtaining four ancient genomes (although most analyses include one, which is common practice in ancient genomics) is undoubtedly an incredible feat in itself, the study does not include many analyses and it's around 3000 words in length. However, it is for the editors to decide whether this is a suitable format for this journal.

>> Dear Reviewer 1,

Thank you for your constructive and thorough comments on our manuscript, we are very pleased you find our study exciting.

We have kept the manuscript as an article as recommended by the Editor.

Please find below our detailed comments to each of the main and minor issues raised.

I have a two main comments:

1) The presentation could be much more exciting, and it lacks context. I think the introduction must mention any known phenotypic/behavioural/environmental differences between the coastal/pelagic ecotypes. A reader would have to refer to the authors' 2021 Science Advances article to get any rationale for why this system is interesting and what is known about it. There is surprisingly little discussion about other systems where ancient genomics has been applied to these kind of questions or directions for future studies.

>> We agree we did not give sufficient context to the study. We further introduced the ecotypes in the introduction and added discussion about other systems.

We added some context on the ecotypes in the introduction L95-100:

“Coastal populations are thought to have been founded by pelagic individuals, and show lower genetic diversity and smaller effective population size than pelagic populations^{10–15}. Morphological differences are found between the two ecotypes but are not consistent across geographical regions^{12,13,16,17}. The coastal ecotype shares behavioural traits across its range such as restricted dispersal, high site fidelity and socially transmitted foraging techniques^{10,18}”

We have also added some discussion about the use of ancient genomes, and explain that our study is fairly novel as we use the ancient sample to shed light on patterns of parallel adaptation, while ancient samples have mostly been used to understand demographic history in non-model wild species. The additional text in the conclusion L376-390 reads as follow:

“Although paleogenomics has been used to understand patterns of adaptation in humans³⁸ and domesticated species³⁹, it has so far mainly been used to understand demographic history in non-model wild species^{40–42} and only in one study to investigate the temporal dynamics of parallel evolution⁴³. In our study, we harness the power of paleogenomics to disentangle the mode of parallel selection through direct observations of the chronology and genetic changes associated with the formation of coastal ecotype populations of bottlenose dolphins. Using subfossil samples pre-dating much of the drift experienced by coastal populations enables us to differentiate drift from coastal adaptation. Overall, our results suggest that coastal variants represent balanced polymorphisms *sensu* Guerrero and Hahn⁴⁴ that were rapidly and repeatedly sieved from standing variation by ecological selection. We thereby provide rare direct evidence of rapid adaptation to newly emerged habitat from standing genetic variation. Our study contributes to the debates of the extent to which parallel evolution is common or rare, and to the role of standing genetic variation on favouring local parallel adaptation^{4,45}. We show that paleogenomics can help solve these debates, and our study can be used as a roadmap for future studies of parallel evolution.”

2) While I’m not an expert in ancient genomics, I believe there are some claims in the manuscript that require more caution or at least better discussion so that the average reader can understand the evidence and believe those claims. These are highlighted in the specific comments below with **.

>> We added more caveats and discussed in greater detail the claims you have highlighted. We detail how we addressed each of issue, highlighted with **, in our comments below.

Specific comments

L55 please mention how many million years ago the mid Holocene was

>> We added “(8,610-5,626 years before present, BP)” after mid-Holocene L55 to refer to the range of ages of our ancient samples.

** L56 I don’t think this claim is clearly discussed or shown in the main text (“We find a

progressive shift from a genetic affinity with pelagic populations towards coastal populations through time'). There are four ancient sequences in this study (three of which are at ultra low coverage). Is this claim in the abstract solely based on the position of the four ancient samples on the PCA? The age of the individuals is not indicated (unless is the shading, but that's not explained in the legend). It is mentioned in the supplementary L511 when describing the single-read PCA approach, but there seems to be no consensus with the other approaches.

>> We give the age of the sample in Table S1. We have now also added the age of the ancient samples in the legend of Figure 1, L147-149: "Calibrated radiocarbon dated age of the ancient samples is as follows in years before present (BP): NMR10326: 8,518-8,346, NMR2273: 7,745-7,572, NMR10151: 7,228-7,036, SP1060: 5,896-5,723."

Our claim is based on the PCAs using various approaches (Figure 1c, Figures S6 and S8) where the oldest sample, NMR10326 (8,610-8,243 years BP), always clusters closest to the pelagic individuals. We noted some reference bias in the analyses with the data mapped to the bottlenose dolphin reference but the oldest sample NMR10326 is still always the closest to the pelagic individuals (Figures S6 and S8). All the samples are "dragged" towards the WNA coastal individuals as the reference genome is from the coastal ecotype of this region.

It is also based on the mitochondrial tree where the oldest sample NMR10326 clusters with pelagic samples from the eastern and western North Atlantic (Figure S2).

We added the age of the sample in the following text in the manuscript about the mitochondrial genome results L164-168:

"The mitogenome sequence of the oldest ancient sample NMR10326 (8,610-8,243 years BP, Figure 1b) clustered with contemporary North Atlantic pelagic haplotypes, while the mitogenome sequences of the three other ancient samples clustered with contemporary Mediterranean and Black Sea samples in a Bayesian phylogenetic analysis (Figure S2)."

We agree with the reviewer that the claim needs better support in the main text and we have added some text in the description of the PCA results L214-223:

"The oldest genome (NMR10326, 8,610-8,243 years BP) clusters with the pelagic populations when mapped to the killer whale reference genome (Figure 1c), while it does not do so when mapping to the bottlenose dolphin reference genome (Figures S6 and S8), likely due to reference bias associated with ancient DNA giving NMR10326 closer affinity on the PCA to the WNA coastal population where the reference genome is from. NMR10326 is however still the closest ancient sample to the pelagic populations (Figures S6 and S8). The clustering of genomic data in PCAs reflects shared mean coalescence times and Identity by State³¹. The position of the ancient genomes suggests that the genetic affinity of ancient samples to North Atlantic coastal populations increases as the age of the samples decreases."

Ref 31: McVean, G. (2009). A genealogical interpretation of principal components analysis. *PLoS genetics*, 5(10), e1000686.

L59 It is important to disclose that the youngest ancient genome was the only one assessed

for presence of coastal-associated genotypes, otherwise it could seem to support the claim above about a shift towards 'coastal' in the four ancient genomes.

>> We have modified the sentence as follow L57-59:

“We assess the youngest genome (5,626 years BP) at sites previously inferred to be under parallel selection to coastal habitats and find it contained coastal-associated genotypes.”

L58 before present should be written in full the first time used in each section, then BP (note L93 also)

>> Corrected, also in the legend of Figure 1.

L90 I think it is very important to define what the authors mean by parallel early in the manuscript, and explain why they chose to refer to linked selection rather than parallel adaptation or evolution.

>> We have clarified what we mean by parallel at the start of the introduction L82-89:

“Parallel adaptation can arise from selection repeatedly acting upon standing genetic variation^{1,2}. However, the timing and independence of selection is still poorly understood for most natural study systems^{3,4}. For example, selection can act upon standing genetic variation independently in multiple derived populations, representing parallel adaptation. It can also act upon standing genetic variation in the ancestral population(s), which then splits into different populations, producing a pattern that can be wrongly inferred as parallel adaptation. Alternatively, standing genetic variation can be shared among derived populations through gene flow, before (i.e. parallel) or after selection (i.e. not parallel)⁵⁻⁹.”

We refer to this process in bottlenose dolphins as “parallel linked selection” rather than “parallel selection” because we cannot be certain that the loci identified in the previous study are the true targets of selection; instead, they may just be physically linked to the true targets of selection. In other words, we can only say with certainty that we have identified the genomic region under selection but not the particular SNPs underlying the phenotypic response. To clarify this, we have added the following sentence L100-106:

“Single Nucleotide Polymorphisms (SNPs), also called variants hereafter, were found to be under parallel selection in geographically distant coastal habitats, that is under both homogenising selection among coastal populations from different ocean basins and divergent selection between coastal and pelagic ecotypes within each ocean basin. We refer to this process as parallel linked selection, as these previous analyses may have identified the targets of selection and/or loci physically linked to these targets¹⁹.”

L94-L95 these two sentences are key to understanding the rationale of the manuscript and I

couldn't understand them. Are the ecotype pairs referring to small populations vs large/widely distributed populations referred to in the second sentence? Otherwise, the 'contrast' between the two sentences isn't clear.

>> We agree the sentences were not clear and we have changed the sentences as follows L100-104: "Single Nucleotide Polymorphisms (SNPs), also called variants hereafter, were found to be under parallel selection in geographically distant coastal habitats, that is under both homogenising selection among coastal populations from different ocean basins and divergent selection between coastal and pelagic ecotypes within each ocean basin."

L97 'ancient ancestry tracts' could be briefly explained for clarity

>> We have added the following text for clarification L106-109: "The coastal-associated variants are found mainly in ancient ancestry tracts in the genomes of coastal dolphins; these genomic regions have a time to the most recent common ancestor between coastal and pelagic counterparts of 0.6 to 2.3 million years BP instead of the genome-wide mean of 0.1 to 0.4 million years BP."

L104 what's a subfossil? And avoid acronyms like ENA when first used (L104), initially thought this referred to the database! L163 WNA also used for the first time in text.

>> The term subfossil refers to bone remains which still contain organic material while fossils are only constituted of inorganic material. The term subfossil is commonly used and we think there should not be any confusion.

We are a bit confused as the acronym was spelt on L104 (L116-117 in the revised manuscript), "the eastern North Atlantic (ENA)".

Thank you for pointing it out for the WNA L163 (L187 in the revised manuscript), we made the correction.

Fig. 1 – could different symbols be used for ancient vs modern in a and b? Additionally, not sure this is a colour friendly palette. I find it particularly difficult to differentiate between the shades of orange ENP/WNA. Perhaps playing with empty vs filled symbols would help.

>> We checked and red/orange and blue are considered to be colour blind friendly colours, and we would like to keep the same colour palette as we used in our study based on modern genomes (Louis et al. 2021) for consistency.

We agree the ancient samples should appear more clearly, and we have now used black triangles instead of green circles. We changed green to black, as green appears as one of the least friendly colours to use.

We have also added the name of the regions for the coastal populations in Figure 1c to help with the difficulty to differentiate between the different shades of orange.

We have also reformatted all the supplementary figures accordingly.

L153 I appreciate the explanations for the different methods used and they seem appropriate.
 >> Thank you.

L162 what analyses is ‘those analyses’ referring to?

>> We changed “those analyses” to “multivariate analyses”.

L173 why is an ancient sample not included in figure s3 PCA? That would be stronger evidence for the lower sequencing depth not pulling samples towards the middle/ towards the ancient genomes.

>> We wanted to test if the positions of the ancient samples on the PCAs were affected by their lower coverage in comparison to higher coverage modern samples, that is why we downsampled a modern individual.

L174 I feel like since the authors went through the effort of doing all these analyses the results of the single read sampling PCAs should be shown in the Supplementary. Isn't this shown in Fig. s8?

>> The results of the single read sampling (SR) analyses are indeed shown in Fig S8 with all modern and ancient samples.

We did not initially show the results of downsampling a modern individual in the SR analyses, but we have now added it as Figure S3b as suggested by the reviewer. As for the smartpca approach, it shows that the ultra-low coverage sample clusters with the rest of the ENAc individuals.

L185 describing patterns along PC2 before PC1 is not common practice, as PC1 explains more variation.

>> We changed the order of the sentences so that PC2 is always after PC1 L210-212:

“Along PC 1, we observe affinity towards the two North Atlantic coastal populations for SP1060 and NMR2273, and to a lesser extent for NMR10151. The ancient samples cluster with the pelagic samples along PC 2 (Figure 1c).”

L237 More basic demographic reconstruction of the modern populations used in this study might help interpreting the admixture graph (PSMC, ROH...). I understand that this has been done before and that's why it is cited, but it could be included here as well for clarity.

>> We have added a summary of the previous results on lines L271-274, noting that we did demographic inference using SMCpp in Louis et al. 2021. It reads as follows:

“Those inferences showed that pelagic populations in different oceanic regions are closer to each other than to their parapatric coastal populations. They also showed that effective population sizes are larger and more stable through time in pelagic than in coastal populations¹⁹.”

We prefer not adding ROH as our ancient samples are too low coverage for those analyses (the method developed for low coverage samples still requires a coverage of 7-8x for ancient samples, ROH in Renaud et al. 2019) and we prefer to focus the present study on analyses in which the ancient samples can bring information on the evolution and parallel adaptation of bottlenose dolphins.

** L257 The claim that the ancient samples are less genetically diverged from pelagic populations (based on the pca of fig 1c) seems tenuous. SP1060 clusters with eastern north Atlantic coastal populations on PC1, and in the many other PCAs presented in the supplementary they tend to be in the middle. The ancient samples might cluster even more clearly with coastal populations if the highly divergent coastal/pelagic pops from the Pacific were removed.

>> The clustering in the PCA is due to the extent of IBS and this will reflect coalescent times, as we have now added in the main text L220-221. For example, when a population goes through founder events much of the genetic variation in future generations will coalesce back to this bottleneck event. This typically leads to descendant samples clustering tightly in PCAs. We see both WNAc and the most northerly cluster of ENAc samples (Scotland) at the extreme right of PC1. The tail of the ENAc is due to more southerly samples that have not undergone the same extent of drift (France and Ireland). We therefore have reworded our sentence as follows L308-313: ‘We observe covariance in alleles among contemporary coastal individuals within a region, but not between contemporary coastal and pelagic individuals, consistent with drift from the pelagic populations in the coastal contemporary individuals. However, based on their position in the PCA space, SP1060 and the other ancient

samples do not share all the drift (and consequent covariance of alleles) experienced by the coastal populations (Figure 1c).”

The results of the PCA are now better discussed earlier L220-223:

“The clustering of genomic data in PCAs reflects shared mean coalescence times and Identity by State ³¹. The position of the ancient genomes suggests that the genetic affinity of ancient samples to North Atlantic coastal populations increases as the age of the samples decreases.”

L261 Please remind the reader here of the dating of the ancient genome (5,979-5,626 BP).

>> We have added the dating.

** L259-263 I’m not sure there is a strong argument for ‘rapid selection’ here. The introduction mentions that the split time of pelagic and coastal ecotypes is 47,800-4,300 years BP (but this was based on few microsats/mitochondrial snps from Louis et al 2014, so perhaps inaccurate), that the coastal habitat in the North Sea emerged 12-6k BP while the ancient genome is dated at 5.8k, so it is on the very recent end of the split time/ after the habitat emerged. The ancient genome is more ‘coastal type’ than pelagic (in every metric that was looked at: PCA, admixture props, D statistics), is it possible that the split happened much earlier, and that this ancient genome represents an early, already coastal-adapted individual? It would help to be reminded of the dates and supporting evidence in this paragraph.

>> We argue that our results do show rapid adaptation. The older ultra-low coverage sample (NMR10326) clearly clusters with the pelagic dolphins both for the mitochondrial and nuclear genomes. The intermediary position of the more recent ancient samples in the PCAs, and the fact they do not share all the drift experienced by modern ENAc individuals (as inferred from PCAs and admixture proportions) from the pelagic populations point towards a more recent divergence from pelagic populations.

As the reviewer highlights, the broad estimate of the split time of 47,800-4,300 years BP is based upon few markers. We feel our new data refines this estimate through more direct dating of our ancient samples. We note that the fossil record shows that the North Sea was mostly an Arctic environment during the deglaciation, and that the temperate marine mammal species only appeared around roughly 8-7,000 years BP (Sorensen et al. 2010, Post et al. 2005). Palaeogeographic models indicate that the southern North Sea got progressively submerged between 9,000 and 6,000 years BP and transitioned from salt marshes to marine waters (Shennan et al. 2016).

We added that the colonisation time was inferred both by genetic studies and fossil record, and that the age of SP1060 is also close to coastal habitat emergence as modelled in Shennan et al. 2006 L317-326:

“SP1060 is dated to 5,979-5,626 years BP, close or shortly after the colonisation time of coastal waters by bottlenose dolphins in the North Sea as indicated by genetic studies ^{12,20}, the subfossil record ^{22,23} and coastal marine habitat emergence around 7-6,000 years BP ²¹. The fossil record shows that the North Sea was mostly an Arctic environment during the

deglaciation, and that temperate marine mammal species appeared around 8-7,000 years BP^{22,23}. Palaeogeographic models indicate that the Southern North Sea was progressively submerged between 9,000 and 6,000 years BP and transitioned from salt marshes to marine waters²¹. This suggests these environmental changes resulted in selection on standing genetic variation with rapid changes in population allele frequencies in newly founded coastal populations.”

We also corrected the timing of the emergence of coastal habitats in the introduction L121-123:

“This coastal habitat emerged when Doggerland was submerged by the North Sea around 8,000 to 6,000 years BP following post-glacial sea-level rise²¹.” We previously indicated 12,000 to 6,000 years BP but, initially, the areas were mainly salt marshes.

L285 I don’t think randomly sampling neutral snps and making trees rules out variation in evolutionary history across different putatively coastal-adaptation genomic regions. The current method only shows the ‘average’ pattern across all outlier snps/genomic regions. In contrast, Lee & Coop style methods make inferences about each region and quantify how many fit each scenario (independent, standing variation, or gene flow origin). This should at least be discussed, especially given that there is clearly gene admixture between modern populations. Additionally, I think adding a panel to Fig. 3 with Fig. S12 could be of value.

>> We have rephrased our text, and acknowledge we cannot use the Lee and Coop method due to the low coverage of the ancient sample L350-361:

“While the Lee and Coop approach is based on trees for each region under selection, we based our analysis on different regions of the genome, thus assuming that the different regions fit the same selection scenarios. We compared the generated tree with ten trees generated using the same number of randomly sampled neutral SNPs. These latter trees do not show a clustering by ecotypes nor strong differences in branch length between coastal and pelagic individuals in contrast to the tree based on the sites under selection (Figure S14). However, we acknowledge that the random sampling of SNPs across the genome may bias towards reconstructing the population consensus tree. Similarly, the topology of the concatenated coastal-associated SNPs may not capture variation in local topologies which are informative of the mode of adaptation. We could not investigate haplotypic differences as in Lee and Coop 2019 due to the lower coverage of the ancient sample, which would impact phasing.”

We have now added a figure with estimates of mean heterozygosity to Figure 3 (figure 3c), and statistical comparison, which is a more robust approach to compare diversity patterns in SP1060, the coastal and pelagic populations than the genotype plot. We have left Figure S12 (now S13) in the supplement.

** L296 what is the support for the claim that there is ‘low drift from pelagic population’ come from? In the legend of Fig. 2 it says that the drift value of the ancient genome is

inflated due to being a single sample and of lower coverage, so how can we know an accurate drift estimate.

>> We have reworded the text L366-373 to

“We find coastal-associated variants in the genome of SP1060, despite it not sharing all the drift experienced by the ENA coastal population after divergence from the pelagic population (Figure 1c). Thus, we hypothesise there was an abundance of coastal-associated standing genetic variation in the ancestral source population; for example, through a large influx of standing genetic variation from coastal populations in refugia, such as the Mediterranean Sea.”

We infer this from the PCA as detailed earlier, we have now added a reference to Figure 1c in the text.

We cannot properly estimate drift in SP1060 using f_4 -statistics in Figure 2 as we only have one sample, and, therefore, all alleles found in SP1060 would be treated as fixed. Singletons and rare alleles in SP1060 that were not shared or were rare at the population level in this ancestral population would be treated as high-frequency/fixed alleles, therefore inflating drift.

We added the following text to the legend L285-288: “This inflation is because all alleles found in SP1060 are treated as fixed, therefore singletons and rare alleles in SP1060 that were not shared or were rare in this ancestral population are treated as high frequency or fixed alleles, inflating estimates of drift along the branch to SP1060”

L299 Perhaps coastal population refugia should be mentioned in the introduction

>> We would prefer to keep this in the discussion as it is only an hypothesis for which we do not have evidence. We added the following sentence to the discussion L371-373: “Habitat models infer the Mediterranean Sea to have been a suitable habitat for bottlenose dolphins during the LGM²⁰.”

L305 ‘we have disentangled neutral and selective processes’ is quite vague and then only a selective process is mentioned in the next sentence, best specify.

>> We now specify that we refer to “coastal adaptation” instead of “selective processes” L382-383: “Using subfossil samples pre-dating much of the drift experienced by coastal populations enables us to differentiate drift from coastal adaptation.”

Reviewer #2 (Remarks to the Author):

Ancient dolphin genomes reveal rapid repeated adaptation to coastal waters

Very interesting and well written manuscript investigating the independent repeated adaptation of bottlenose dolphins to coastal habitats using contemporary and subfossil genomes.

What are the noteworthy results?

The authors used a combination of 60 contemporary and 4 subfossil genomes to provide insights on the chronology and independent parallel selection in the formation of coastal bottlenose dolphins. The authors conclude that coastal bottlenose dolphin populations have evolved rapidly and independently through parallel selection from standing genetic variation found in ancestral pelagic populations, thus providing evidence of rapid adaptation to newly emerged habitat.

Will the work be of significance to the field and related fields? How does it compare to the established literature? If the work is not original, please provide relevant references.

Although the divergence of coastal bottlenose dolphins from pelagic populations has been investigated in previous literature, this article is the first one to use subfossil samples to explore the genetic changes associated with the formation of coastal bottlenose dolphins, providing unique genomic data and robust evidence of parallel independent selection. These results are of great importance to our understanding of bottlenose dolphin population dynamics and they represent a significant contribution in evolutionary biology by providing rare evidence of rapid adaptation to new habitats from standing genetic variation.

Does the work support the conclusions and claims, or is additional evidence needed?

I think the data and analysis supports very well the conclusions presented in the manuscript. One of the highlights and what makes in my opinion the manuscript robust, is that the authors used subfossil genomes from a period when bottlenose dolphins were starting to colonize new coastal habitats following post-glacial sea-rise, providing strong evidence for the claims proposed in the article.

Are there any flaws in the data analysis, interpretation and conclusions? Do these prohibit publication or require revision?

Is the methodology sound? Does the work meet the expected standards in your field?

I did not find any flaws in the data analysis, interpretation or conclusions. However, I have not worked using ancient samples before and I have not performed many of the analyses used in the manuscript. Nevertheless, from my point of view the data analysis is very rigorous and scientifically sound. I believe the manuscript meets the standards and would be a great contribution to evolutionary biology in general.

Is there enough detail provided in the methods for the work to be reproduced?

Yes, the authors provide a comprehensive section in materials and methods and all the supplementary data needed to access the genomes used in the study.

>> Dear Reviewer 2,

Thank you for your comments on our manuscript, we are very pleased you enjoyed our manuscript.

Reviewer #3 (Remarks to the Author):

Dear authors,

In my opinion your paper represent an important case study on how to incorporate paleogenomic data to address the origin and evolution of populations within a marine mammal, one that appears to have convergent evolution at the population level. The results are quite interesting, and tie together your detailed analyses with potentially related ecological changes linked to sea level rise since the last glacial maximum. That said, the results themselves will probably be of main interest to expert that work on delphinids (and maybe other cetaceans), whereas the methods used to come to your conclusions may be of broader interest.

>> Dear Reviewer 3,

Thank you for your constructive and thorough comments on our manuscript, we are very pleased you find our study important.

We think that the results on parallel selection and the pace of local adaptation are of interest to evolutionary biologists in general regardless of the species on which they work. As also suggested by reviewer 1, we now better highlight the novelty of our study in the use of ancient samples to address parallel evolution and broader impact in the conclusion L376-390:

“Although paleogenomics has been used to understand patterns of adaptation in humans³⁸ and domesticated species³⁹, it has so far mainly been used to understand demographic history in non-model wild species⁴⁰⁻⁴² and only in one study to investigate the temporal dynamics of parallel evolution⁴³. In our study, we harness the power of paleogenomics to disentangle the mode of parallel selection through direct observations of the chronology and genetic changes associated with the formation of coastal ecotype populations of bottlenose dolphins. Using subfossil samples pre-dating much of the drift experienced by coastal populations enables us to differentiate drift from coastal adaptation. Overall, our results suggest that coastal variants represent balanced polymorphisms *sensu* Guerrero and Hahn⁴⁴ that were rapidly and repeatedly sieved from standing variation by ecological selection. We thereby provide rare direct evidence of rapid adaptation to newly emerged habitat from standing genetic variation. Our study contributes to the debates of the extent to which parallel evolution is common or rare, and to the role of standing genetic variation on favouring local parallel adaptation^{4,45}. We show that paleogenomics can help solve these debates, and our study can be used as a roadmap for future studies of parallel evolution.”

The present study appears to differ from previous work from the lead and other authors in its incorporation of subfossil data as well as using methods and models that allow for different hypotheses to be tested that account for apparent convergent molecular evolution in coastal populations of *Tursiops truncatus*.

My main criticism of your approach is the change in sampling of individuals in the analyses that infer gene flow/divergence between present and past populations of *T. truncatus*. In the Bayesian tree (S2) extant pelagic and coastal populations are intermingled with Black Sea and Mediterranean individuals/populations (although I am unsure if any of these represent coastal populations). Thus it would seem appropriate to include these populations in subsequent analyses. I am guessing that their exclusion is related to the assumption that the divergence of coastal populations represented in the Northeast Atlantic occurred in that region during the past 20K years, and that the Mediterranean and Black Sea Populations are peripheral to this question. If so, that basis for sampling in the different analyses could be better explained.

>> We were only able to include Black Sea and Mediterranean Sea individuals in the mitochondrial genome analyses (Bayesian tree in Figure S2) as we used publicly available data for this analysis (Table S3 gives the accession numbers). There is no available whole genome resequencing data for the Black Sea and Mediterranean Sea bottlenose dolphins. Due to the lack of available whole genome resequencing data, we do not know for sure the ecotype of those individuals, although they are believed to be coastal in these two regions. We added this information in the legend, and we clarified that those are publicly available data in the legend of figure S2. The text in the legend reads: “Individuals from the eastern Mediterranean Sea and Black Sea are believed to be coastal ¹⁶. New mitochondrial genome haplotypes have been assembled from data from Louis *et al.* (2021) ⁶ and we also used publicly available mitochondrial genome haplotypes from Nykänen *et al.* (2019) ²⁵ and Moura *et al.* (2013) ¹⁶.”

The text in the supplementary material reads:

“Mitochondrial genomes of *T. truncatus*, extracted from carbon-dated ancient subfossils (n = 4), contemporary samples (n = 70; including the 57 samples from Louis *et al.* (2021), three newly sequenced ones plus ten samples from Nykänen *et al.* (2019), see supplementary data file), or downloaded from GenBank (n = 68, see Table S3 and supplementary data file) ¹⁶ were first aligned in software MEGA X ²⁴ with a *T. aduncus* reference mitogenome (KF570335.1) downloaded from GenBank.”

On a related note, I think your paper could be improved if you better focused on the unique insights provided by the subfossil genomic data. Could the inclusion of extant Mediterranean dolphins (in place of the subfossil SP1060) have yielded essentially identical results?

>> We do focus on the unique insights provided by the subfossil genomic data. We only include analyses where the subfossil(s) can add information. Specifically, how these samples

provide unique resolution to the chronology and tempo of genetic changes associated with adaptation and drift.

At present the relationship between any ancestral Mediterranean population and the ENAc population remains hypothetical as there is no available whole genome re-sequencing data for the Mediterranean Sea. A present-day Mediterranean dolphin genome would have several thousands of years of evolutionary changes through drift and selection along the branch from the MRCA with the ENAc population to the present-day Mediterranean dolphins (**Figure A**). This TMRCA would also likely be older than the TRMCA between SP1060 and any ENAc dolphin. The subfossil SP1060 is geographically proximate to the ENA coastal population and is close to being directly ancestral to the ENA coastal population. Thus, the best resolution of the evolutionary processes through time comes from including a temporally sampled genome(s).

Figure A. Schematic representation of the possible relationships between ENAc, SP1060 and Mediterranean common bottlenose dolphins.

I found it noteworthy that figure S11 (which excluded fossil data) is very similar to fig. 2 (which does).

>> Regarding Figure 2 and S11, we showed that when including modern and called genotypes (Figure S11), we get a similar graph topology as when including the ancient sample and pseudo-haploid data (Figure 2). This shows the robustness of our approach using pseudo-haploid genomes. We feel the ancient sample may be too low coverage to include in an admixture graph with called genotypes. It is relatively standard practice in studies using ancient DNA to compare admixture graph results with pseudo-haploid and called genotypes, see for example Ramos-Madrigal et al. 2021 Current Biology. Thus, the fact that figure S2 with the ancient sample is very similar to figure S11 supports our approach. If they were not concordant, we would be concerned of bias either in the called genotyped or pseudo-haploid calls.

We acknowledge that the reasoning behind running the analysis using both pseudo-haploid data and the ancient samples, and with called genotypes and the modern populations only was not clear from what we included in the main text; thus we have rephrased the text as follow L251-255:

“Graphs were estimated using pseudo-haploid data mapped to the killer whale reference genome and using the killer whale as an outgroup. We find similar topologies when using called genotypes for the contemporary populations only and mapping to the bottlenose dolphin reference genome, indicating that the approach based on pseudo-haploid data is robust (Figure S11).”

We have also added the following text to better highlight why including SP1060 is important for our inferences L301-306:

“We have shown that due to its relationship with contemporary populations, the ancient sample SP1060 can provide a unique temporal resolution to the chronology and pace of genetic changes associated with the colonisation of coastal habitat in the eastern North Atlantic. Thus, we investigated and compared patterns of variation in SP1060 and contemporary populations at sites previously identified as evolving under parallel linked selection ¹⁹.”

In general, I think your paper could be improved if it showed how a statistical model of fit (or difference between competing evolutionary trees of populations) becomes stronger/more clear cut with the addition of the fossil data. No doubt the acquisition and sequencing of subfossil data required substantial effort on your part, and clearly showing the impact of such data might help encourage other teams to collect such data for their clade/population of interest.

>> Please see our response above regarding the insights provided by the ancient sample(s).

The objective of the admixture graph is to better understand the relationship of the ancient sample and the present-day modern populations. These results help us to better understand the mode and chronology of adaptation, which are the objectives of our study. To be clear, our goal is not to achieve the best fit admixture graph possible, it is to understand the relationships of the ancient sample SP1060 to our present-day samples. In particular, whether sample SP1060 provides a useful proxy for the ancestral population of the present day ENA coastal modern populations.

As detailed above we have now better explained on L251-255 that we also run the graph analyses on the modern populations only using called genotypes to check we get the same results as with the pseudo-haploid data, to ensure our results are robust.

Further, we used qpbrute which tests across all possible histories to find the “best” admixture graphs with no outlier f -statistics (i.e. all $|Z|$ were <3) for both the pseudo-haploid data with SP1060 and the genotype data without SP1060. This is essentially a test of the goodness of fit of the data to the proposed model.

As mentioned earlier, we also now better highlight the broader implications of the work as suggested by reviewer 1 in the conclusion L376-390:

“Although paleogenomics has been used to understand patterns of adaptation in humans³⁸ and domesticated species³⁹, it has so far mainly been used to understand demographic history in non-model wild species⁴⁰⁻⁴² and only in one study to investigate the temporal dynamics of parallel evolution⁴³. In our study, we harness the power of paleogenomics to disentangle the mode of parallel selection through direct observations of the chronology and genetic changes associated with the formation of coastal ecotype populations of bottlenose dolphins. Using subfossil samples pre-dating much of the drift experienced by coastal populations enables us to differentiate drift from coastal adaptation. Overall, our results suggest that coastal variants represent balanced polymorphisms *sensu* Guerrero and Hahn⁴⁴ that were rapidly and repeatedly sieved from standing variation by ecological selection. We thereby provide rare direct evidence of rapid adaptation to newly emerged habitat from standing genetic variation. Our study contributes to the debates of the extent to which parallel evolution is common or rare, and to the role of standing genetic variation on favouring local parallel adaptation^{4,45}. We show that paleogenomics can help solve these debates, and our study can be used as a roadmap for future studies of parallel evolution.”

Other Comments

1) Largely citing previous work, the divergence of the Northeast Atlantic coastal populations of *T. truncatus* is correlated with the rise in sea level after the last glacial maximum and the flooding of Doggerland by the North Sea. While this is certainly possible, there are always coastal habitats regardless of the position of sea level. What changes is the complexity and type of coastal habitats (e.g. estuaries are associated with flooding of river valleys) and the total area of shallow marine environments (depending on the position of sea level relative to the shelf break). A few more sentences explaining why this rise in sea level is expected/predicted to have caused a change in selection would be helpful.

>> We have added a few sentences in the introduction L126-129 to explain why the changes in condition following rise of sea level may have led to coastal adaptation:

“The emergence of new coastal habitat, with different abiotic and biotic environmental conditions such as salinity, depth and prey species, may have affected the physiology, foraging strategies and behaviour of bottlenose dolphins and created opportunities for local adaptation to occur.”

We corrected the timing of the emergence of coastal habit in the introduction L121-123:

“This coastal habitat emerged when Doggerland was submerged by the North Sea around 8,000 to 6,000 years BP following post-glacial sea-level rise²¹.” We previously indicated 12,000 to 6,000 years BP but at first the areas were mainly salt marshes.

We also note that the fossil record shows that the North Sea was mostly an Arctic/sub-Arctic environment during/following the deglaciation, and that the temperate marine mammal species only appeared around roughly 8-7,000 years BP. Palaeogeographic habitat models

indicate that the southern North Sea got progressively submerged between 9,000 and 6,000 years BP and transitioned from salt marshes to marine waters (Shennan et al. 2016) .

We have now added that the colonisation time was inferred both by genetic studies and fossil record, and that SP1060 is also close to coastal habitat emergence as modelled in Shennan et al. 2006 L317-326:

“SP1060 is dated to 5,979-5,626 years BP, close or shortly after the colonisation time of coastal waters by bottlenose dolphins in the North Sea as indicated by genetic studies ^{12,20}, the subfossil record ^{22,23} and coastal marine habitat emergence around 7-6,000 years BP ²¹. The fossil record shows that the North Sea was mostly an Arctic environment during the deglaciation, and that temperate marine mammal species appeared around 8-7,000 years BP ^{22,23}. Palaeogeographic models indicate that the Southern North Sea was progressively submerged between 9,000 and 6,000 years BP and transitioned from salt marshes to marine waters ²¹. This suggests these environmental changes resulted in selection on standing genetic variation with rapid changes in population allele frequencies in newly founded coastal populations.”

2) in Fig. S11, a, b, and c are not explained. Is one of these hypotheses more favored than the other?

>> The following text is in the Supplementary material: “The Bayes Factors showed a non-significant support of the first model in comparison to the other “mirror” two, which have similar log likelihoods (Bayes Factor of 0.88). We increased the chains to four millions iterations with a burn-in of 3 millions, but it did not help in discriminating between the three graphs.”

We added the following text to the legend of Figure S11:

“The Bayes Factors showed a non-significant support for the first model in comparison to the other “mirror” two, which have similar log likelihoods (Bayes Factor of 0.88).”

Reviewer #4 (Remarks to the Author):

This is a nice straightforward analysis providing an interesting empirical look at parallel evolution in coastal dolphin populations. The methodology is quite robust and the findings are well supported. I have a few minor comments to improve clarity and statistical robustness. It is fundamentally appropriate for Nat Comms with a few edits. Thanks to the authors for an interesting read.

>> Dear Reviewer 4 ,

Thank you for your constructive and thorough comments on our manuscript, we are very pleased you find our study interesting.

211–213: This is a bit confusing, the text implies comparison between ENA and WNA coastal populations regarding the ancient sample, but the d-stat form indicates pelagic samples. Is the finding that with ENA vs WNA in the “coastal” position, ENA has the stronger non-zero d-statistic? Is that with the same pelagic sample(s)? That seems to be the case but could be clarified.

>> Yes you are right we mean that ENAc has a stronger non-zero d-statistics than WNAc. We have clarified it in the text as follows L241-244:

“Accordingly, SP1060 shares a higher excess of derived alleles with the ENA coastal population than with the WNA coastal population, in statistics of the form $D(\text{coastal,pelagic}; \text{SP1060}, \text{Orca})$ as shown by a stronger non-zero D -statistic when the coastal individual is from the ENA than from the WNA.”

We refer to the comparison as follows: $D(\text{ENAc}, \text{ENAp}; \text{SP1060}, \text{Orca})$ and $D(\text{WNAc}, \text{WNAp}; \text{SP1060}, \text{Orca})$, so with the pelagic samples from the same region. We also estimated $D(\text{ENAc}, \text{WNAp}; \text{SP1060}, \text{Orca})$ and the value is even more negative (see Figure S9). We estimated $D(\text{ENAp}, \text{WNAc}; \text{SP1060}, \text{Orca})$, which is positive, and therefore $D(\text{WNAc}, \text{ENAp}; \text{SP1060}, \text{Orca})$ would be the same value but negative. The statistics show stronger non-zero D -statistics in all cases where ENAc is included. The clarification in the sentence above should cover all these four combinations.

Figure 2 – it would be useful to provide at least one z-score to demonstrate the quality of the fit. At the least, please define “outlier f -statistics” in the main text (presumably $|Z| > 3$ as in the supplement). Also this figure is a bit sparse. I would consider annotating it further with some of the description from the main text interpretations.

>> We have added the information both in the main and figure legend, that no outlier f -statistics means that all branches had $|Z| < 3$. The strongest outlier was -0.96, we have added the following text L248-251: “Testing across all possible histories, we find one admixture graph with no outlier f -statistics (i.e. all $|Z|$ were < 3 , the greatest deviation from 0 was -0.96) using qpBrute (Figure 2), which explores the space of all possible admixture graphs of a given maximum complexity, under a brute-force approach^{35,36}.”

We have added text to the legend of Figure 2:

Figure 2. Evolutionary relationships between the ancient individual SP1060 (5,979-5,626 years BP) and the North Atlantic contemporary bottlenose dolphin populations as inferred using qpBrute. Solid arrows indicate the relationships between populations/samples and the numbers at their right side correspond to the estimated genetic drift represented by the arrow. This graph was the only one of all possible graph combinations presenting no outlier f -statistics (i.e. all $|Z|$ were < 3). Populations include eastern North Atlantic coastal (ENAc) and pelagic (ENAp) populations, and western North Atlantic coastal (WNAc) and pelagic (WNAp) populations, and the outgroup is the killer whale (KW). Note that the drift value for SP1060 is inflated due to being a single and lower-coverage sample. This inflation is because all alleles found in SP1060 are treated as fixed, therefore singletons and rare alleles in SP1060 that were

not shared or were rare in this ancestral population are treated as high frequency or fixed alleles, inflating estimates of drift along the branch to SP1060. Dashed lines are admixture edges and the arrows indicate the inferred direction of admixture, with the numbers reflecting the percentage of ancestry deriving from each lineage. The graph reveals a first split between the lineage that gave rise to the contemporary coastal populations and SP1060, and the lineage that gave rise to the majority of ancestry in contemporary pelagic populations. The WNA coastal and SP1060 are shown as independent lineages. The ENA coastal is depicted as a clade whose ancestry is a mixture of the two ancestral groups leading to SP1060 and WNA coastal. Ancient sample SP1060 is therefore not a direct ancestor of the contemporary ENA coastal dolphins. The ancestry of both North Atlantic contemporary pelagic populations appears to be an admixture of approximately 30% of the lineage giving rise to the coastal populations, and approximately 70% from a deeply divergent lineage.

248: Can you please explicate further and/or provide a citation for why lower coverage and a single sample would artifactually inflate the inferred drift parameters?

>> We added the following text to the legend L285-288: “This inflation is because all alleles found in SP1060 are treated as fixed, therefore singletons and rare alleles in SP1060 that were not shared or were rare in this ancestral population are treated as high frequency or fixed alleles, inflating estimates of drift along the branch to SP1060.”

257: Figure 1c isn't totally clear on the fact that SP1060 (the one actually included in this analysis) is substantially less diverged from the pelagic populations compared with ENAc. I would generally prefer that position on a PCA be used as a hypothesis and then tested with explicit statistical methods. Figure 1d seems to indicate the pattern more clearly. Additionally, a quick f_4 statistic in the form $f_4((SP1060, ENAc), pelagic, OG)$ should be clarifying as to whether SP1060 is more pelagic-like than is ENAc.

>> We have now added text to explain how we interpret the PCA patterns in the main text L220-223: “The clustering of genomic data in PCAs reflects shared mean coalescence times and Identity by State³¹. The position of the ancient genomes suggests that the genetic affinity of ancient samples to North Atlantic coastal populations increases as the age of the samples decreases.”

We see both WNAc and the most northerly cluster of ENAc samples (Scotland) at the extreme right of PC1. The tail of the ENAc is due to more southerly samples in the ENA that have not undergone the same extent of drift (France and Ireland).

We have reworded our sentence at previously L257 as follows L308-313: “We observe covariance in alleles among contemporary coastal individuals within a region, but not between contemporary coastal and pelagic individuals, consistent with drift from the pelagic populations in the coastal contemporary individuals. However, based on their position in the PCA space, SP1060 and the other ancient samples do not share all the drift (and consequent covariance of alleles) experienced by the coastal populations (Figure 1c).”

We have run D-statistics as they are more appropriate for single samples. The D-statistic (((ENAc,SP1060),pelagic),Orca) is significantly negative indicating that the ENAc is closer to ENAp or WNAp than SP1060 is. However, D-statistics are also influenced by patterns of gene flow, and this result is likely due to gene flow between ENAc and ENAp as inferred in Figure 2.

260: Heterozygosity needs to be more robustly established to make this statement. With a low-coverage sample, depending on the specific strategy used, sequencing error could more easily affect called genotypes than in high-coverage modern samples. In fact, there is no information that I can find in the supplement about the generation of the called genotypes, do they derive from the Louis et al 2021 methodology? This is very important to include. The heterozygosity estimation also needs to be done very carefully in light of potential coverage-based artifacts linked to sequencing error. The visual of S12 as cited is not sufficient to quantitatively demonstrate higher heterozygosity in the ancient sample.

>> Yes we used the same methodology as in Louis et al. 2021. That is, we filtered the bam files for excessive coverage and sites which were out of HWE and negative in all populations to remove paralogues, etc.

We agree the methodology needed more details and we expanded it in the following text in the supplement:

“Then, we kept only the mapped reads with a mapping quality of at least 25. We removed repeated regions as identified using RepeatMasker ²⁷, regions of excessive coverage and mapping artefacts (sites which were out of Hardy Weinberg Equilibrium with negative inbreeding coefficients in all populations as they could be paralogues or other mapping artefacts), and the sex chromosomes using a combination of bedtools v. 2.25.0 and samtools (see details in Louis et al. 2021⁶). We also removed all the scaffolds which were shorter than 10 Mbp.”

Please, note that this text was already in the supplement:

“We generated a vcf file including the contemporary individuals and SP1060 following the filtering steps as in Louis *et al.* (2021) ⁶. In addition, in the vcf file, we kept only sites with data with 90% of the individuals and sites with no missing data in the ancient sample.”

We also added some text in the main text to clarify our approach L179-181: “Data processing and filtering followed Louis et al. 2021 ¹⁹ with some modifications specific to ancient DNA data such as recalibration base quality scores according to DNA damage patterns, which are detailed in the supplementary material.”

We also recalibrated the base quality scores according to damage patterns for the ancient samples, as included in the sentence above.

We have now also estimated heterozygosity in *angsd* for the SNPs under parallel linked selection; *angsd* is well suited to analyse low coverage data as it can use genotype likelihoods instead of called genotypes. We have added the figure as Figure 3c and Figure S12. Heterozygosity (H_e) is significantly higher in coastal than pelagic populations in all regions (all $p < 0.01$), and mean H_e for SP1060 is within the range of the values found in the ENAc population but not ENAp.

We have added a reference to this figure in the statement about heterozygosity L315-317: “The ancient genome shows excess heterozygosity as found in coastal individuals ¹⁹, despite its lower coverage (Figures 3c, S12 and S13).”

Figure 3. Patterns of genetic variation of the SNPs under repeated selection to coastal habitat in contemporary common bottlenose dolphin individuals and ancient individual SP1060. These include 2,122 SNPs with no missing data in SP1060 out of the 7,165 SNPs identified in Louis et al. (2021) ¹⁹, mapped to the bottlenose dolphin reference genome. Populations include coastal and pelagic ecotypes from the eastern North Atlantic (ENAc and ENAp), western North Atlantic (WNAc and WNAp) and eastern North Pacific (ENPc and ENPp). a) Principal component analysis including Eigen values; the proportion of genetic variance captured by each component is indicated in the axes; and (b) Neighbour-joining tree showing the genetic structure of the common bottlenose dolphin samples for this particular SNP set, with grey circles indicating bootstrap node support values higher than 95%. c) Heterozygosity (H_e) estimated for all the sites under parallel linked selection for each population using *angsd* 0.921; the violin plots indicate the kernel probability density of the data, the box indicates the interquartile range, and the horizontal marker indicates the median

of the data. There is only one data point for SP1060. Heterozygosity is significantly higher in coastal than pelagic populations within all geographical regions ($p < 0.01$ for all pairwise comparisons). Note that genome-wide coastal populations have less diversity than pelagic populations as described in Louis et al. (2021)¹⁹. SP1060 shows a mean H_e of 0.295 which is very close to the mean of 0.291 for ENAc. The same figure with all the data points can be found in Figure S12b. d) Illustrations of the three possible scenarios of parallel and non-parallel selection as per Lee and Coop 2019⁴. The star represents a beneficial mutation and the brown bars the period of time during which selection acted upon standing genetic variation. Scenario i) highlights standing genetic variation originating in the ancestral population and then being targeted by selection independently in each derived population. Scenario ii) corresponds to a scenario where standing genetic variation originated in one derived coastal population and was then shared through gene flow with the other coastal populations. Subsequently selection acted independently in each coastal population to increase the frequency of the adaptive variants. Scenarios i) and ii) represent parallel selection. Scenario iii) shows selection acting upon standing genetic variation in a shared ancestral population. This does not fit parallel selection due to the non-independence of adaptive allele frequencies in the derived populations.

268: This sentence is quite confusing, although the meaning becomes clear through the next 2 scenarios. The most useful approach would be to illustrate these three scenarios schematically.

>> As per this suggestion, we have illustrated the three scenarios schematically and added the figure to figure 3.

Figure 3d. Illustrations of the three possible scenarios of parallel and non-parallel selection as per Lee and Coop 2019⁴. The star represents a beneficial mutation and the brown bars the period of time during which selection acted upon standing genetic variation. Scenario i) highlights standing genetic variation originating in the ancestral population and then being targeted by selection independently in each derived population. Scenario ii) corresponds to a scenario where standing genetic variation originated in one derived coastal population and was then shared through gene flow with the other coastal populations. Subsequently selection acted independently in each coastal population to increase the frequency of the adaptive variants. Scenarios i) and ii) represent parallel selection. Scenario iii) shows selection acting

upon standing genetic variation in a shared ancestral population. This does not fit parallel selection due to the non-independence of adaptive allele frequencies in the derived populations.

We have also reworded the sentence as follows L330-333:

“Selection would be considered independent if it acted upon standing genetic variation in each derived population (scenario i, Figure 3d)⁴, meaning selection would act along branches that are not shared among the two North Atlantic coastal populations and SP1060 in Figure 2.”

276: The visual in fig S12 is very difficult to interpret in light of the three scenarios presented. It’s possible to eyeball it and detect some kind of clustering, but this isn’t a terribly robust approach.

>> We have now added a figure with estimates of heterozygosity for all populations and SP1060, including statistical comparison, to complement figure S12 (now S13), the new figure is figure 3c and figure S12.

280-281: This contradicts 3b, which if interpreted in a hierarchical clustering or phylogenetic sense, shows that SP1060 is basal to WNAC with ENAC as the more divergent group. Please clarify, and again, statistical assessment of affinity to groups would be preferable to interpreting the visual in 3a.

>> We would like to stress that the NJ tree is a consensus tree not representing one evolutionary history. The ancestral node of the branch ending with SP1060 at its tip is also basal to a node which leads to both ENA and WNA samples, as we see in Figure 3a where SP1060 is intermediary between ENA and WNA in the PCA space. We have clarified the sentence as follow L343-345:

“SP1060 clusters most closely with two individuals of the ENA coastal population (Figure 3a), but diverges close to the basal node of all the Atlantic coastal populations (Figure 3b).”

To add a statistical assessment of affinity to groups, we have performed 100 bootstraps of the neighbour-joining tree and indicate with grey circles nodes where bootstrap support is higher than 95%. We have done the same for the tree with the neutral SNPs in the supplementary material (Figure S14).

306: “balanced” slightly implies balancing selection, which is not in evidence. Consider rephrasing.

>> We added “sensu Guerrero and Hahn⁴⁵” to balanced polymorphism to clarify.

Reviewers' Comments:

Reviewer #1:

Remarks to the Author:

The authors have carefully reviewed my comments and provided many clarifications on the results section. The improved figures now aid the interpretation of the results. I am glad the authors have added some more introduction and discussion, making it more appealing to a broader audience and specifically highlighting what the novelty is. I think this study is an exciting use of paleogenomics to shed light on the mode and tempo of parallel adaptation to the environment, and is ready for publication in its current form.

Reviewer #3:

Remarks to the Author:

I think you have done an excellent job responding to my comments as well as those of the other reviewers. In particular I appreciate the text added to explain why this particular study would be of broad interest, not just to those studying marine mammals.

Reviewer #4:

Remarks to the Author:

The authors have answered my concerns satisfactorily, and the MS is suitable for acceptance.

REVIEWERS' COMMENTS

Reviewer #1 (Remarks to the Author):

The authors have carefully reviewed my comments and provided many clarifications on the results section. The improved figures now aid the interpretation of the results. I am glad the authors have added some more introduction and discussion, making it more appealing to a broader audience and specifically highlighting what the novelty is. I think this study is an exciting use of paleogenomics to shed light on the mode and tempo of parallel adaptation to the environment, and is ready for publication in its current form.

Dear reviewer 1,

We are happy to hear that you are pleased with our revisions and find our study exciting.

On behalf of the co-authors,

Marie

Reviewer #3 (Remarks to the Author):

I think you have done an excellent job responding to my comments as well as those of the other reviewers. In particular I appreciate the text added to explain why this particular study would be of broad interest, not just to those studying marine mammals.

Dear reviewer 3,

We are happy to hear that you are pleased with how we have addressed your and the other reviewers' comments.

On behalf of the co-authors,

Marie

Reviewer #4 (Remarks to the Author):

The authors have answered my concerns satisfactorily, and the MS is suitable for acceptance.

Dear reviewer 4,

We are happy to hear that you are satisfied with our revisions.

On behalf of the co-authors,

Marie